# Engineering intelligent chassis cells *via* recombinase-based MEMORY circuits

Brian D. Huang [1,2], Dowan Kim [1,2], Yongjoon Yu [1] & Corey J. Wilson [1] ✉

Synthetic biologists seek to engineer intelligent living systems capable of decision-making, communication, and memory. Separate technologies exist for each tenet of intelligence; however, the unification of all three properties in a living system has not been achieved. Here, we engineer completely intelligent *Escherichia coli* strains that harbor six orthogonal and inducible genome-integrated recombinases, forming Molecularly Encoded Memory via an Orthogonal Recombinase arraY (MEMORY). MEMORY chassis cells facilitate intelligence via the discrete multi-input regulation of recombinase functions enabling inheritable DNA inversions, deletions, and genomic insertions. MEMORY cells can achieve programmable and permanent gain (or loss) of functions extrachromosomally or from a specific genomic locus, without the loss or modification of the MEMORY platform – enabling the sequential programming and reprogramming of DNA circuits within the cell. We demonstrate all three tenets of intelligence via a probiotic (Nissle 1917) MEMORY strain capable of information exchange with the gastrointestinal commensal *Bacteroides thetaiotaomicron*.

Synthetic biologists have developed separate technologies to emulate decision-making[1–9], intercellular communication[10–16], and the equivalent of memory[17–25] in myriad chassis cells. We posited that all three properties can be unified in a single chassis cell to form an intelligent synthetic biological system (see Fig. 1, Supplementary Note 1). We hypothesized that an intelligent chassis cell could be engineered via the coordination and optimization of orthogonal recombinase functions mapped to discrete biosensing operations. Recombinases (large serine integrases) are enzymes that mediate site-specific DNA inversion, excision, and insertion events depending on the orientation of cognate attachment (*att*) sites[26,27]. Several recombinases have been identified and experimentally characterized, enabling said catalysts to be repurposed for use in synthetic genetic circuits[20,28–31]. One of the key advantages of serine integrase function is that this type of recombination can be deployed in prokaryotic[19,30] and eukaryotic[23,32] systems. In addition, studies have demonstrated that recombinase function can be artificially regulated by way of inducible promoters[18,19,22]. These works utilized well-characterized transcription factors (TFs) to regulate recombinase expression, allowing at most three orthogonal

inducible (plasmid-based) recombinases to be deployed in a single *E. coli* chassis cell[22,30]. Additional studies have used similar strategies to create recombinase circuits incorporating more relevant biosensors for targeted applications[33–35], including the detection of key biomarkers for non-invasive diagnostics[36–38] – similarly, these iterations of recombinase circuits utilized fewer than three regulated recombination operations.

In a recent study, we demonstrated that the regulation of recombination can be achieved via synthetic transcription factors (i.e., synthetic repressors and synthetic anti-repressors) and can be deployed concurrently with Transcriptional Programming (T-Pro)[39] – see Supplementary Note 2. T-Pro[4,5,40,41] has emerged alongside Cello circuit design software[2,6,42] as promising technologies for engineering cellular decision-making (i.e., tenet 1, see Fig. 1). Here we sought to engineer an iteration of synthetic memory (tenet 2) that is concurrently compatible with both Marionette[9] and T-Pro transcription factors. We posited that the envisioned engineered cells would enable the development of bespoke living programs capable of executing unified decision-making, communication, and memory.

[1]Georgia Institute of Technology, School of Chemical & Biomolecular Engineering, 311 Ferst Drive, Atlanta, GA 30332-0100, Georgia. [2]These authors contributed equally: Brian D. Huang, Dowan Kim. ✉e-mail: corey.wilson@chbe.gatech.edu

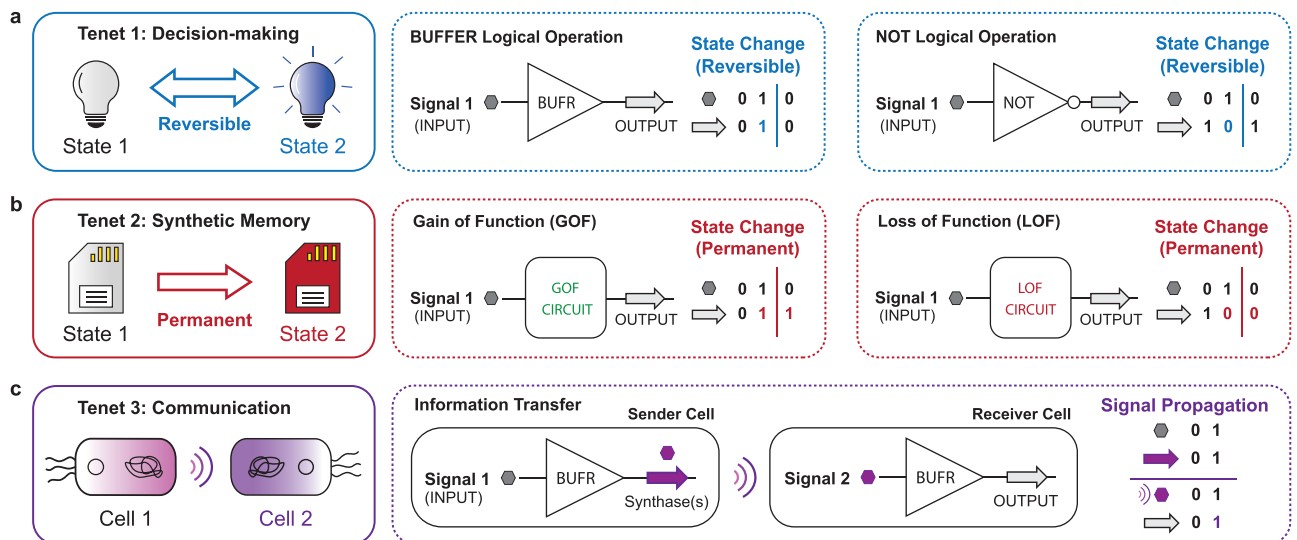

**Fig. 1 | Components of an intelligent biological system. a** The concept of decision-making is illustrated (left). Transient state changes are achieved by controlling gene expression with transcription factor-based gene circuits to achieve Boolean logic (middle, right). **b** The concept of synthetic memory is illustrated (left). Permanent state changes are achieved through the manipulation of genetic components at the DNA level. Transiently expressed recombinases facilitate gain-of-function (middle) or loss-of-function (right) via genetic circuits. **c** The concept of communication is illustrated (left). Information transfer is achieved through inducible small-molecule production and subsequent sensing by sender and receiver cells, respectively (right).

In this work, we develop *Escherichia coli* chassis cells with a genomically integrated memory array composed of six orthogonal, inducible recombinases – regulated by a set of transcription factors commonly used in the Marionette biosensing array (i.e., PhlF, TetR, AraC, CymR, VanR, and LuxR). The expression level of each recombinase is carefully optimized to achieve near digital switching of cell genotype when induced to perform a specific recombination function. We develop 24 fundamental gain-of-function (GOF) and loss-of-function (LOF) memory circuits for both inversion and excision attachment site configurations. To expand the capacity of memory functions we develop a means of CRISPR-Cas9-mediated protection of recombinase action (CRISPRp). Namely, we show that catalytically inactive *Streptococcus pyogenes* Cas9 (dCas9) can be directed to a given recombinase attachment site and successfully prevent recombination with high (~99%) efficiency. Moreover, we demonstrate that CRISPRp of a given *att* site can be programmed with fundamental decision-making via T-Pro transcription factors – with concurrent MEMORY operation. In addition, CRISPRp is used to develop a next-generation recombinase-based state machine (ngRSM) to demonstrate an application of this post-translational control mechanism. Finally, we demonstrate how our engineered chassis cells can be used to program information exchange between a probiotic *E. coli* Nissle with a transplanted MEMORY platform and the commensal bacterium *Bacteroides thetaiotaomicron* – i.e., a chassis cell that we previously demonstrated is capable of supporting the full range of two-input Transcriptional Programs via ligands that can be used as dietary supplements[41]. The pairing of a non-colonizing probiotic strain with *B. thetaiotaomicron* establishes an important platform technology, defining the next generation of consortium-based living therapeutics capable of concurrently supporting all three tenets of intelligence (Fig. 1).

## Results

### Engineering an array of inducible recombinases

There have been at most three independently inducible recombinases deployed in a single *E. coli* cell[22,30]. We sought to increase this number by identifying six putatively orthogonal recombinases (A118, Bxb1, Int3, Int5, Int8, and Int12) from the large serine integrase family that have been previously characterized and used in synthetic biology applications[18–20,22]. Next, we identified six TFs (PhlF, TetR, AraC, CymR, VanR, and LuxR) that have been rigorously optimized and shown to be orthogonal to one another from the Marionette biosensing array[9]. For each recombinase, we arbitrarily assigned a regulating TF to control enzyme expression (Fig. 2). Genetic libraries were created for each recombinase to determine the optimal expression levels that would result in: (i) minimal leakiness in the uninduced state, and (ii) high recombination efficiency in the context of inversion upon induction of the regulating promoters (Fig. 2a, b). Each recombinase library consisted of an inducible promoter, a degenerate ribosome binding site (RBS) sequence[43], a degenerate start codon, and two degradation tags of variable strength. These libraries were cloned into a single-copy bacterial artificial chromosome (BAC) with the intent of mimicking genomic expression levels[44]. To test the function of each inducible recombinase, output circuits were designed where a strong, inverted promoter ($P_{J23119}$) was flanked by anti-aligned *att* sites followed by a green fluorescent protein (*gfp*) gene, harbored on a low-copy (3–5 copy) pSC101 plasmid (Fig. 2a). We designated this genetic circuit architecture as an inversion GOF circuit. Note, in this study we are using the iteration of GOF and LOF defined by Huang et al.[41]., which utilized an inert reporter output (e.g., green fluorescent protein) as a proxy for function.

We initially used the Marionette-Wild[9] strain of *E. coli* to screen the resulting libraries, as this strain has the six TF regulators integrated into its genome. Each recombinase library was co-transformed with the corresponding inversion GOF plasmid, and transformants were randomly screened using a memory assay that we developed (Methods). Briefly, transformants harboring a recombinase circuit were grown in M9 minimal medium (MM) with and without the cognate inducer, and after a defined growth period transferred into fresh MM without inducer. These final cultures were then analyzed using flow cytometry to assess levels of recombination. This assay ensured that the expression state of the cells being analyzed was dependent on the inducer input history, rather than the current growth environment. After screening our libraries and recharacterizing promising clones, we were able to isolate variants that exhibited low recombination activity without inducer, complemented by high levels of recombination when transiently induced – determined by flow cytometry (Fig. 2a, b, and Supplementary Fig. 1a).

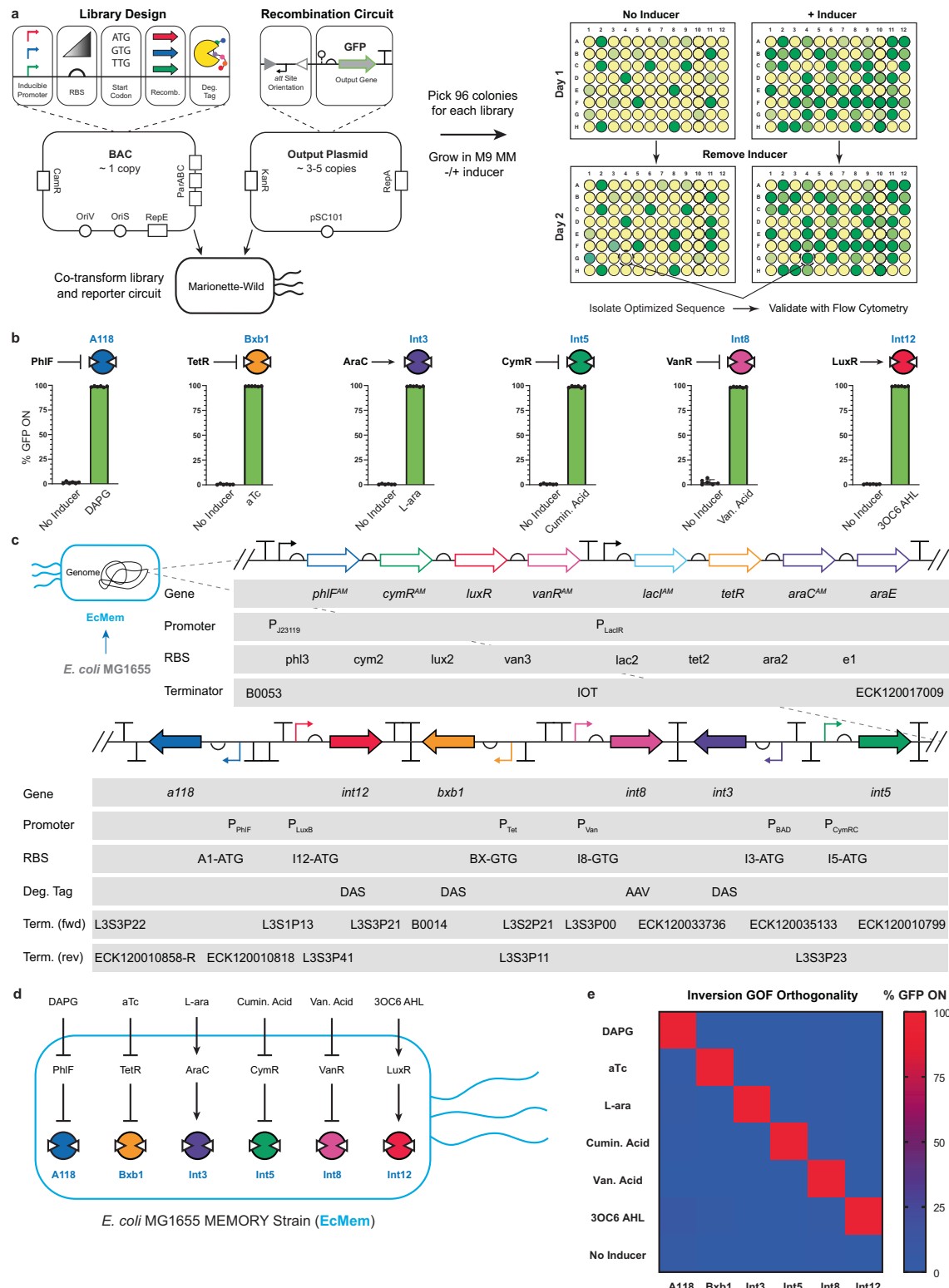

## Coordinating insulated recombinase expression from a genomic locus

Once the expression level for each recombinase was optimized, we sought to integrate the six inducible cassettes into the genome of Marionette *E. coli* MG1655. The majority of prokaryotic recombinase circuits reported to date have utilized medium- and high-copy plasmids to harbor the recombinases[17–20,22,30,38,45], with the exception of a genome-integrated system reported in *Bacteroides thetaiotaomicron*[46].

Several independent investigations have demonstrated that multicopy genetic circuits can impose a significant resource burden on the host[47–57], leading researchers to move toward the construction of single-copy genetic circuits[44,58,59]. There are several advantages to creating single-copy expression systems, including enhanced genetic stability and reduced risk of horizontal gene transfer[58,60–62]. To this end, we designed our recombinase expression system to be implemented at the single-copy level at the outset of this study.

**Fig. 2 | Engineering the MEMORY platform. a** The variables involved in library generation for recombinase expression levels are shown. Libraries contain randomized combinations of an inducible promoter, a degenerate RBS sequence, a variable start codon, and a C-terminal degradation tag. Each recombinase library is co-transformed with a reporter plasmid containing an inverted promoter upstream of the green fluorescent protein (*gfp*) gene. Individual transformants were screened for low levels of recombination (uninduced) and high levels of recombination (induced). **b** The performances of isolated optimized clones from (**a**) are shown. % GFP ON denotes the percentage of cells expressing GFP as measured by flow cytometry (See Methods). **c** The complete genetic schematic of the recombinase expression system is shown. The transcription factors are those reported in Ref. 9

unaltered. The recombinase genes are inserted directly downstream of the ECK120017009 terminator. Full sequences of parts are given in Supplementary Data 1. The full sequence of the recombinase expression cluster is given in Supplementary Data 2. **d** A map of ligand input to unique recombinase output via transcription factor-regulated expression is shown for EcMem. **e** Orthogonality between the six recombinases is shown. The EcMem strain transformed with each of the six inversion GOF circuits is assayed with all single inducers. The heatmap shows the percentage of cells with the reporter circuit recombined. Data in (**b**) and (**e**) represent the average of $n = 6$ biological replicates, with groups of three taken on two separate days. Error bars in (**b**) correspond to the SEM of these measurements. Source data are provided as a Source Data file.

Prior to genomic integration, we used the BAC as a testbed for the design of an insulated locus for the six recombinases. We anticipated that the induction of the promoter of one recombinase could cause transcriptional readthrough of a second recombinase coding sequence in the same reading frame. To diminish this possibility, we incorporated strong terminators[63] upstream and downstream of each recombinase, as well as alternated the direction of transcription of each successive gene to provide further transcriptional insulation. We cloned an initial version of this insulated locus into the BAC and used Marionette-Wild to perform the memory assay with each of the six inversion GOF reporter plasmids, however, this time using all sets of inducers for each of the six circuits to assess for cross-induction of recombinases. Despite this initial insulation attempt, we still saw evidence of transcriptional readthrough and cryptic promoter activity causing unintended activation of certain recombinases (Supplementary Note 3, also see Supplementary Fig. 2). These effects were largely mitigated with the strategic insertion of additional terminators in the recombinase expression array. After these modifications, the only instance of significant cross-induction was observed with A118 *att* sites unexpectedly recombining (i.e., ~9%) in the presence of the 3OC6 AHL inducer corresponding to the Int12 recombinase (Supplementary Fig. 3).

After the recombinase expression system showed suitable orthogonality when harbored on the BAC, we proceeded to integrate the insulated set of genes into the Marionette MG1655 genome (see Fig. 2c, Methods, Supplementary Data 1, and Supplementary Data 2). We integrated the sequence immediately downstream of the regulating TFs, simultaneously removing the unused TFs harbored in the Marionette-Wild genome with the exception of LacI (to be used later for orthogonal proof-of-concept T-Pro regulation, unifying tenet 1 and tenet 2). Herein, we refer to this genetic construct as the Molecularly Encoded Memory via an Orthogonal Recombinase arraY (MEMORY) platform – establishing tenet 2 – and we designated the new strain of *E. coli* as EcMem (Fig. 2d). We then repeated the orthogonality experiment to confirm that each genome-integrated recombinase performed equivalently to its BAC counterpart (Fig. 2e). Each recombinase successfully recombined its inversion GOF target with >97% efficiency while exhibiting <3% recombination in the uninduced state. In addition, we characterized the relative rate of recombination for each recombinase by measuring the amount of time required for recombination under ideal growth conditions in minimal medium with inducer (Fig. 3a). Each genome-integrated recombinase was tested for its recombination rate using the inversion GOF circuit as well as an excision-based version of GOF synthetic memory. All recombinases showed complete recombination after approximately 12 h when maintained in exponential growth.

### Expanding synthetic memory with additional recombinase circuits

After the development and characterization of the EcMem strain using inversion GOF circuits, we then expanded our memory system by designing, building, and testing 18 additional synthetic memory circuits (Fig. 3b–e). Namely, we sought to design a second GOF circuit

based on DNA excision, as well as two LOF circuits based on either DNA inversion or excision. We posited that engineering a diverse set of optimized (i.e., near digital) circuits capable of both inversion and excision would allow for the accelerated design and development of complex genetic programs that utilize the strategic arrangement of *att* sites (i.e., nesting) that rearrange transcriptionally regulating elements such as promoters, terminators, or non-coding RNAs. To this end, we developed the complementary inversion LOF circuit by flanking an in-frame promoter with anti-aligned *att* sites upstream of the *gfp* gene (Fig. 3c). The excision GOF circuit was designed to have a strong constitutive promoter upstream of aligned *att* sites flanking two terminators in series, followed by the *gfp* gene (Fig. 3d). Finally, the excision LOF circuit was designed by flanking an in-frame constitutive promoter with aligned *att* sites, followed by the *gfp* gene (Fig. 3e).

When designing recombinase circuits, several orientations of *att* sites were tested for each recombinase given that cryptic promoters and terminators can arise from the *att* sequences[30,45] (Supplementary Fig. 4). We tested all 18 additional synthetic memory circuits using the EcMem strain and demonstrated that each memory operation performed *on par* with the initial inversion GOF circuits (Fig. 3c–e, also see Supplementary Note 4). Interestingly, when these circuits were characterized using single recombinases harbored on the BAC, the performance was less efficient when compared to the genome-integrated recombinases (Supplementary Fig. 1). We posited that this was due to increased variability in initial recombinase expression levels when cells were co-transformed with a pSC101 circuit plasmid and a BAC, as there may be a delay in the genome-integrated TFs initiating the regulation of a recombinase's promoter. This served to highlight another advantage of the genome-integrated recombinase array, where the recombinase expression levels are already minimized due to the TFs having achieved stable concentrations prior to the introduction of a pSC101 circuit.

### Robustness and resource burden of the recombinase array

To assess the long-term utility and genetic stability of the integrated memory system, we performed an extended growth experiment with the EcMem chassis cell harboring the inversion GOF circuits. Specifically, we transformed the EcMem strain with each of the six inversion GOF plasmids and grew these cells for 11 days (~200 doublings) – (Fig. 3f). The cultures were passaged every 12 h into fresh minimal medium without inducers for the entire experiment and analyzed by flow cytometry. Additionally, every other day a separate set of cultures was inoculated, containing the cognate inducer to activate the corresponding GOF circuit. A standard memory assay was performed on these induced cultures to assess for genomic maintenance of the recombinase array and potential loss of recombinase function through genetic drift. All recombinases performed consistently throughout the entire 11-day period with negligible recombination occurring in the absence of inducers. Each attempt to induce the recombinases was successful with >95% recombination efficiency (Fig. 3f, also see Supplementary Fig. 5). This experiment demonstrated that the integrated recombinase array is stable against evolution and that memory circuits can be maintained successfully for greater than 10 days. Furthermore,

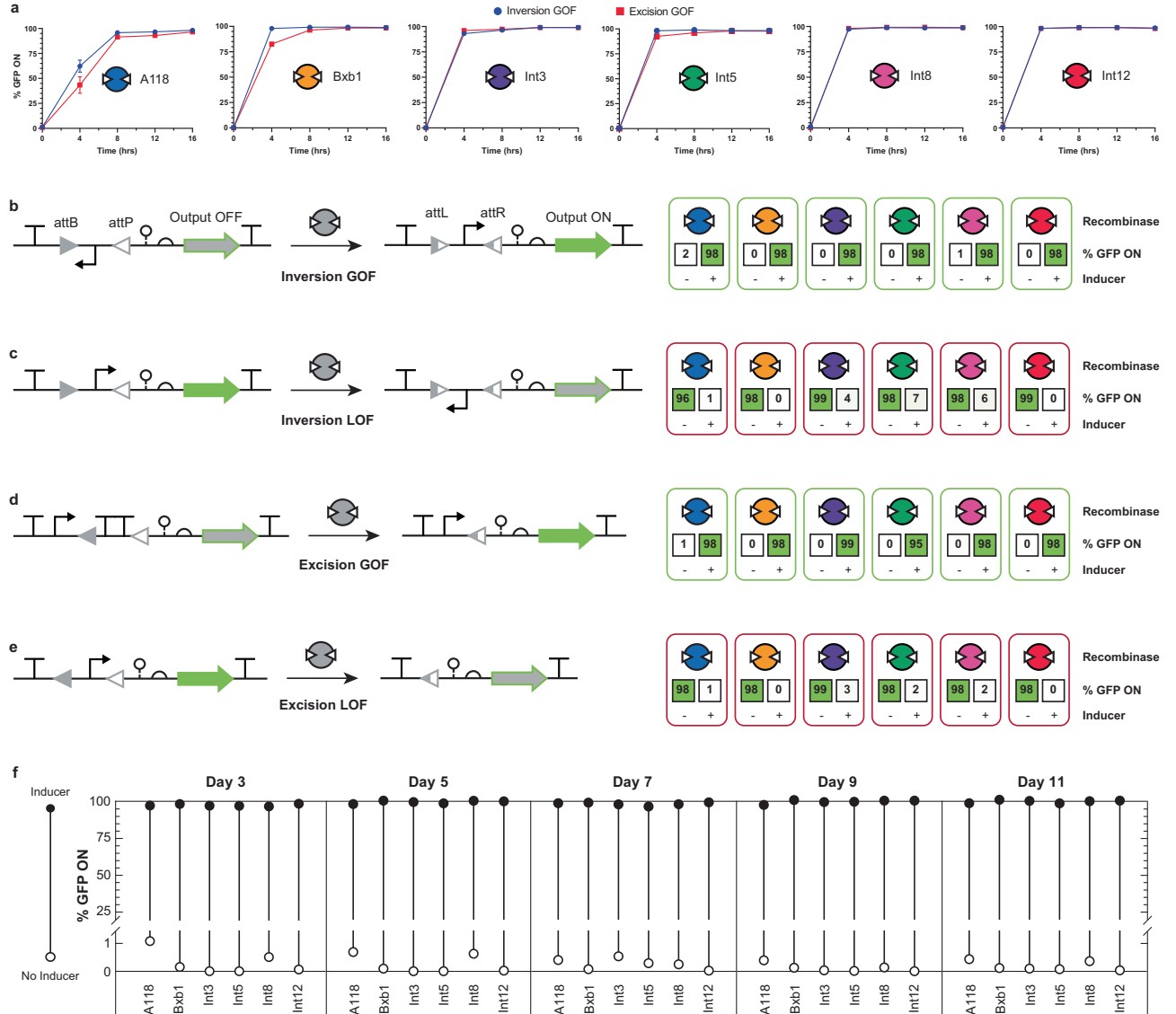

**Fig. 3 | Characterization of the MEMORY platform. a** Kinetics of recombinase activity is shown. The time required for recombination is shown for the EcMem strain transformed with each of the six inversion GOF circuits (blue circles) and each of the six excision GOF circuits (red squares). **b** The inversion GOF circuit architecture is shown (left) along with the performance of 6 unique circuits corresponding to the 6 recombinases (right). **c** The inversion LOF circuit architecture is shown (left) along with the performance of 6 unique circuits corresponding to the 6 recombinases (right). **d** The excision GOF circuit architecture is shown (left) along with the performance of 6 unique circuits corresponding to the 6 recombinases (right). **e** The excision LOF circuit architecture is shown (left) along with the performance of 6 unique circuits corresponding to the 6 recombinases (right).

**f** Genetic stability of the MEMORY platform is shown. The EcMem strain transformed with each of the six inversion GOF circuits is cultured continuously for 11 days. Every other day the cultures are used to seed media with inducers to assess for maintenance of recombinase functionality. Open circles represent no inducer, filled circles represent induced cultures. A single biological time course is shown. See Methods for additional information and Supplementary Fig. 5 for an additional biological replicate. Data in (**a-e**) represent the average of $n = 6$ biological replicates, with groups of three taken on two separate days. Error bars correspond to the SEM of these measurements. All data represent experiments performed using the EcMem strain. Source data are provided as a Source Data file.

we analyzed the growth rate of EcMem during recombinase expression to assess potential toxicity and metabolic burden. Even when all six recombinases were induced simultaneously, we saw minimal impact on cell growth rate (Supplementary Fig. 6).

## Extrachromosomal MEMORY programming, erasing, and re-programming

In principle, the EcMem chassis cell can be used to execute bespoke memory programs supplied on extrachromosomal DNA or via genome-integrated circuits. Here we aimed to demonstrate MEMORY programming by way of extrachromosomal (plasmid DNA) circuits. We posited that we could design functional memory circuits with an

additional feature that would enable the complete removal of the extrachromosomal DNA at any point on cue – effectively erasing the plasmid-based circuit – while retaining the genomically integrated MEMORY platform. A putative synthetic memory eraser was designed via an aligned pair of *att* sites flanking the origin of replication of the pSC101 plasmid. As designed, the addition of the L-arabinose inducer should result in the origin of replication being excised, preventing extrachromosomal DNA propagation – i.e., erasing the corresponding memory circuit, and resetting the EcMem chassis cell (Supplementary Fig. 7). Here, the erasable DNA plasmid contained a two-output memory circuit, designed for independent inducible GOF by way of the expression of GFP or mKate – via transient exposure to small

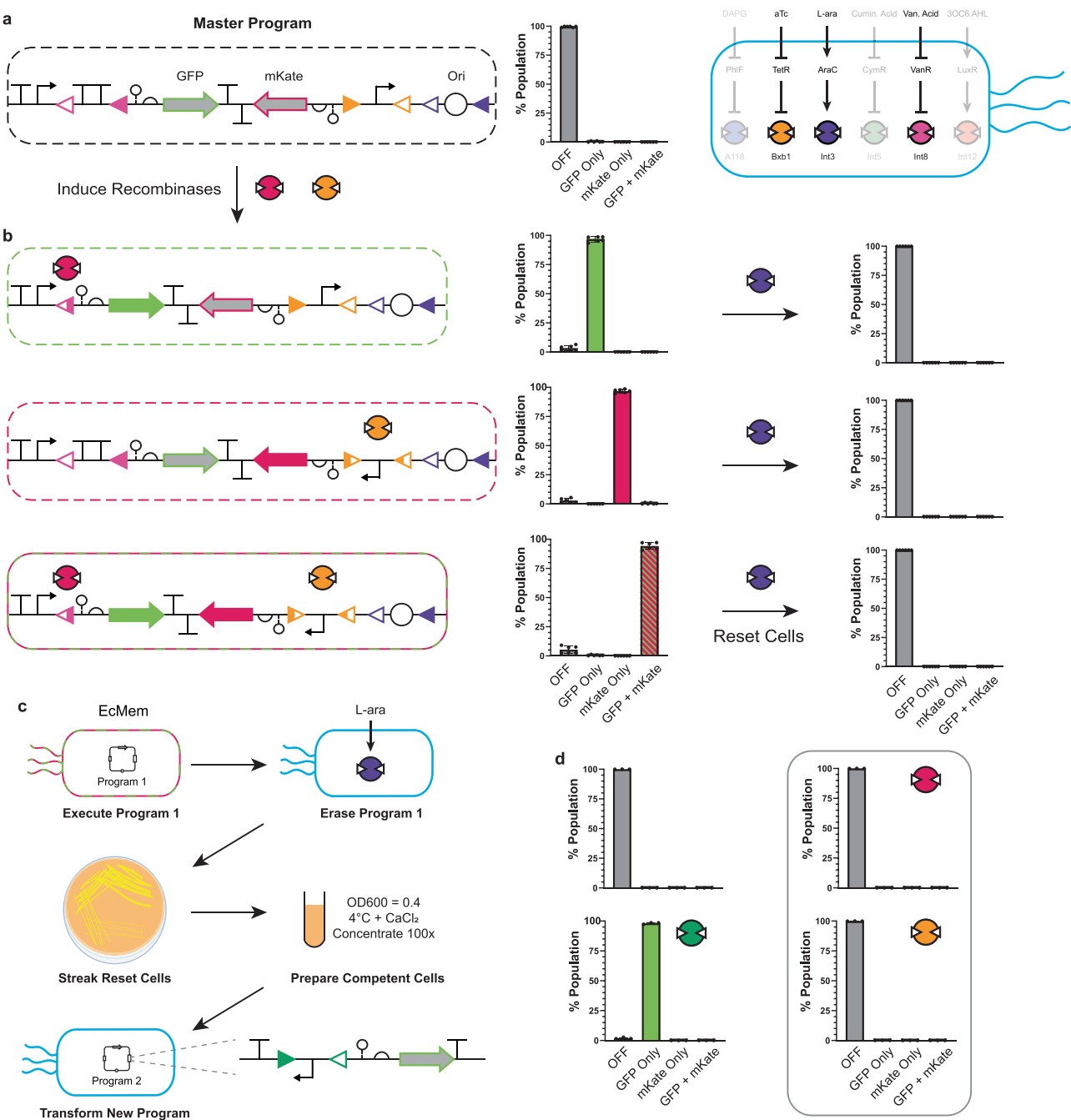

**Fig. 4 | A multi-input multi-output program with a cellular reset. a** A four-state program is shown (left). The Int8 excision GOF circuit controls the activation of GFP, the Bxb1 inversion GOF circuit controls the activation of a red fluorescent protein (mKate), and Int3 controls the excision of the pSC101 origin of replication. The distribution of cells in the four possible states of fluorescent protein expression after growth in medium without inducers is shown (middle). The recombinases involved in the program are highlighted in EcMem (right). **b** The program behavior is shown when Int8 is induced (top), Bxb1 is induced (middle), or when both recombinases are induced (bottom). The percentages of fluorescent cells are shown to the right of each circled DNA arrangement. After initial program activation, cells were grown with L-arabinose to induce excision of the plasmid origin and resetting of cellular genotype (right bar graphs). **c** Testing of the origin eraser

fidelity is shown. Following program execution and erasing (**a**, **b**), cells were made chemically competent and transformed with a new program (the Int5 GOF circuit). **d** Fidelity of the cellular reset is shown. Reset cells from the "GFP + mKate" state in (**b**) were transformed with the Int5 inversion GOF circuit. The distributions of cells in each expression state are shown when induced with cuminic acid, vanillic acid, and aTc to demonstrate consistent strain performance with a new program, and loss of the previous program. For (**a**, **b**), data represent the average of *n* = 6 biological replicates, with groups of three taken on two separate days. For (**d**), data represent the average of *n* = 3 biological replicates from a single experiment. Error bars correspond to the SEM of these measurements. All data represent experiments performed using the EcMem strain. Source data are provided as a Source Data file.

molecules vanillic acid or aTc, respectively (Fig. 4a). We demonstrated that we could induce each GOF proxy individually or simultaneously to achieve the desired fluorescent protein output(s) and subsequently reset the MEMORY cells by way of origin excision (Fig. 4b, also see

Supplementary Fig. 8). We estimated that greater than 99.9% of cells were successfully reset (i.e., lost the pSC101 plasmid) by flow cytometry analysis. After demonstrating that we could erase the two-output extrachromosomal circuit, we showed that the reset MEMORY

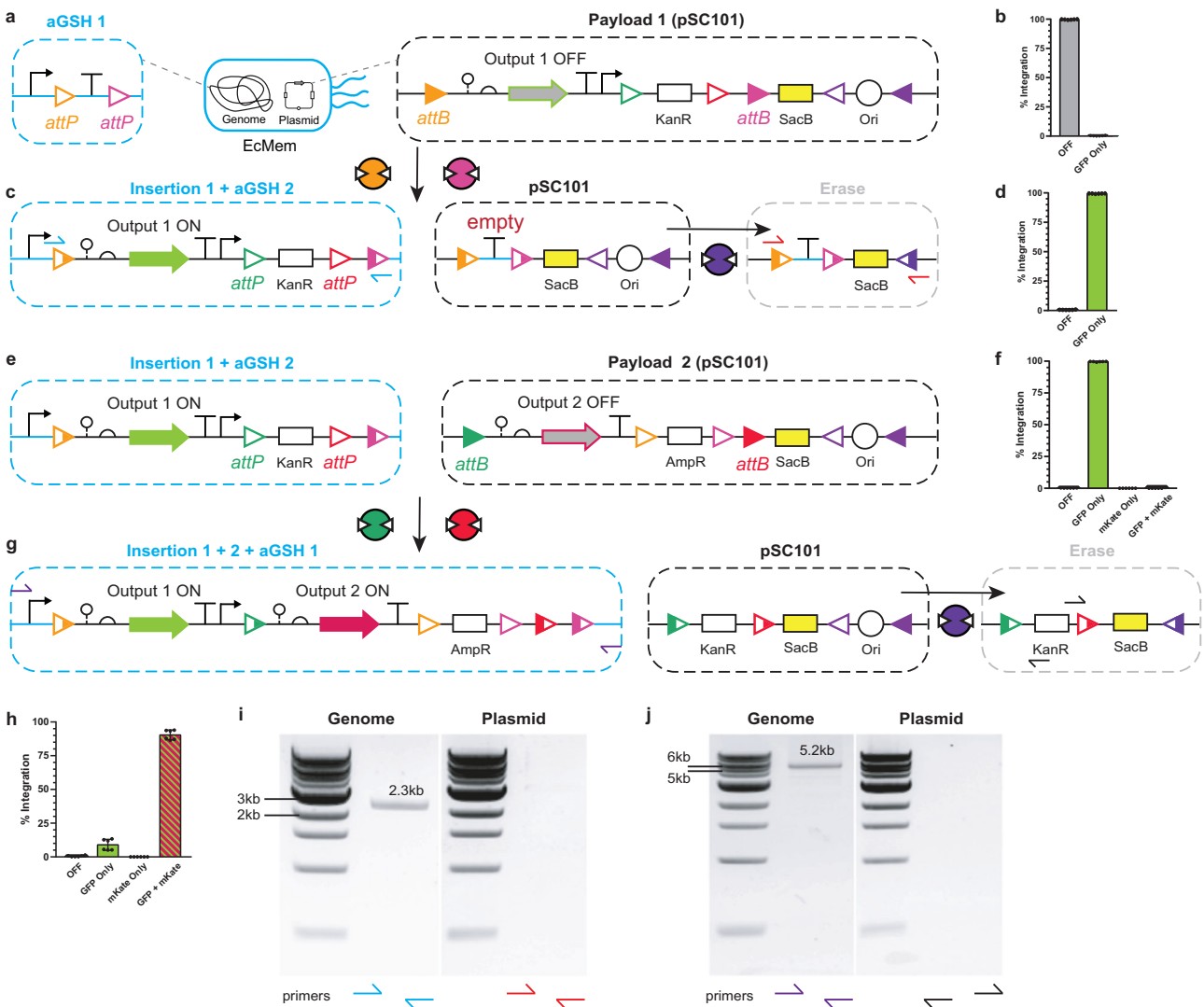

**Fig. 5 | MEMORY recording through genomic integration. a** The first genomic safe harbor (aGSH1) recording circuit (left) and payload 1 (right) are shown. **b** The percentage of cells expressing GFP prior to recombinase induction is shown. **c** The recombined genomic DNA (left) and plasmid DNA (middle) states are shown after induction of Bxb1 and Int8, followed by Int3 induction to erase the plasmids (right). **d** The percentage of cells expressing GFP after recombinase induction is shown, representing the percentage of integration. **e** The new genomic sequence and aGSH2 (left) are shown with payload 2 (right). **f** The percentage of cells expressing GFP and mKate prior to recombinase induction is shown. **g** The recombined genomic DNA (left) and plasmid DNA (middle) states are shown after induction of Int5 and Int12, followed by Int3 induction (right). **h** The percentage of cells expressing GFP and mKate after recombinase induction is shown. **i** Representative colony PCR products from cells in (**c**) are shown. Primers are denoted by colored half arrows. **j** Representative colony PCR products from cells in (**g**) are shown. Data in (**b, d, f**), and (**h**) represent the average of $n = 6$ biological replicates, with groups of three taken on two separate days. Error bars correspond to the SEM of these measurements. Source data are provided as a Source Data file.

cell could be transformed with a new circuit and maintain the genome-integrated array, and execute a new extrachromosomal program (Fig. 4c, d). Specifically, we selected a single reset colony and made the cells chemically competent to transform the Int5 inversion GOF circuit. We proceeded to grow these cells with and without the Int5 inducer (cuminic acid), as well as the two relevant inducers from the original circuit (vanillic acid and aTc). Consistent with our expectation, the circuit was faithfully executed in response to the Int5 inducer and showed no response to the other inducers (Fig. 4d).

**Programmed DNA insertion by way of MEMORY chassis cells for genome engineering**
In addition to DNA inversion and excision circuits, we also developed proof-of-concept integration circuits that allow for the inducible (programmed) insertion of genetic elements into the genome of EcMem chassis cells (Fig. 5). Park et al. introduced a robust iteration

of recombinase-based genomic insertion technology that leverages single *att* site landing pads[58]. However, given that this landing pad technology is based on a single *att* site payload insertion, the initial recombination event incorporates the entire plasmid. To remove any unwanted DNA the authors used the FLP recombinase and cognate attachment sites to minimize the footprint of the insert. In this landing pad system, the recombinases are supplied via plasmid DNA and are unregulated. Likewise, Santos et al. demonstrated in an earlier study that the Cre recombinase can be used to insert a specific DNA fragment into the *E. coli* genome using two sets of orthogonal *att* sites – again unregulated[64]. Here, we introduce the next iteration of recombinase-based genomic insertion technology that leverages our MEMORY platform for the programmed insertion of DNA into the genome of EcMem chassis cells. In principle, MEMORY chassis cells can accomplish the programmed genomic insertion of an exact payload in a single step – i.e., without the need to remove unwanted

integrated DNA – and can support programmed serial genomic integrations.

To accomplish programmed integration, we created an artificial genomic safe harbor (aGSH) using a nonsynonymous pair of *attP* sites corresponding to two recombinases (Methods). In principle, the aGSH can be integrated with genetic information stored on a plasmid between a set of complementary *attB* sites (Fig. 5a). To demonstrate MEMORY-facilitated programmed insertion, we positioned a promoter upstream of the aGSH in the genome of the EcMem chassis cell and paired the engineered safe harbor with a set of complementary *attB* sites directing a payload (*Payload 1*) containing the *gfp* gene and a kanamycin resistance gene (*kanR*) flanked by a second pair of non-synonymous *attP* sites – creating a new aGSH upon genomic integration (Fig. 5a–c). Additionally, the pSC101 donor vector was equipped with the memory eraser to remove the "empty" plasmid after integration, as well as the *sacB* gene to provide a counterselection and eliminate integrants receiving the entire plasmid. In turn, we used flow cytometry to quantify the efficiency of integration after insertion and plasmid erasing, but before SacB counterselection, at ~99% by measuring the fluorescence of GFP (Fig. 5d). We then confirmed the integration via colony PCR of the inserted region (Fig. 5c,i) after plating cells on LB agar with sucrose (Methods). We then demonstrated sequential integration by inserting the *mKate* gene (*Payload 2*) in the second aGSH, simultaneously reintroducing the first aGSH flanking an ampicillin resistance gene (Fig. 5e–g). The percentage of cells that received the second integration was shown to be ~90% based on cytometry, and was confirmed via colony PCR (Fig. 5h,j, also see Supplementary Fig. 9). We also tested a version of this integration circuit where the pSC101 plasmid did not contain the *sacB* gene, and observed nearly identical performance (Supplementary Fig. 9). These results demonstrated that our recombinase system can be used for efficient genomic integration, which should allow for the rapid development of derivative strains for user-specific applications.

## MEMORY expansion via programmable CRISPR protection

Serine integrases will bind to and recombine all pairs of cognate *att* sites concurrently, with the capability of recombining orthogonal sites defined by unique dinucleotide (core) sequences. Consequently, to achieve two or more discrete (decoupled) memory operations requires the use of multiple recombinases – i.e., one recombinase per set(s) of cognate attachment sites. Accordingly, we sought to develop a technology that would allow for the systematic expansion of single recombinase circuit capacity. Our goal was to program one recombinase to independently recombine two or more cognate attachment sites, including the ability to differentiate between identical *att* sites that would result in unique recombination events. We posited that dCas9 could be repurposed to bind (within or in proximity to) specific recombinase *att* sites and protect the DNA from programmed recombination. Shur and Murray presented a proof-of-concept of unregulated dCas9-mediated protection of a single *att* site cognate to Bxb1 in a cell-free (TX-TL) environment[65], which is distinct from canonical CRISPR interference (CRISPRi) of transcription[66]. We posited that we could achieve an advance over this design via (i) generalization of dCas9-mediated *att* site protection to a panoply of recombinases in a living system, (ii) demonstration that dCas9-mediated protection can be controlled in parallel with MEMORY, and (iii) demonstration that dCas9-mediated protection is amenable to Transcriptional Programming.

To test our assertion, we designed a program where dCas9 was constitutively expressed from a BAC (also harboring an inducible recombinase), and the inversion GOF circuit was modified to include an IPTG-inducible single guide RNA (sgRNA) targeting a single *att* site (Fig. 6a). We devised three protection targeting strategies based on anti-aligned (inversion) *att* site architectures, with the goal of leveraging natural and synthetic PAM (protospacer adjacent motif) sites to

guide dCas9 (Fig. 6b, c). We then validated that our method of dCas9-mediated protection could successfully prevent recombination with >95% efficiency for all six recombinases represented in our MEMORY system (Fig. 6d, also see Supplementary Figs. 10 and 11, and Supplementary Note 5). Additionally, we verified that we could control dCas9-mediated protection with synthetic TFs from our Transcriptional Programming (i.e., decision-making) toolkit (Fig. 6e, f, also see Supplementary Fig. 12) enabling us to unify tenets 1 and 2 in a single chassis cell. Namely, we demonstrated that three of our synthetic TFs – that constitute fundamental BUFFER operations – could be used to form simple programs that run orthogonally to MEMORY inputs. Herein, we refer to our programmable iteration of dCas9-mediated protection of DNA as CRISPR protection (CRISPRp). Note, we have demonstrated in previous work that the development of BUFFER operations using our synthetic TFs correlates with the ability to construct complex Boolean decision-making[41]. Collectively, this result demonstrates that EcMem chassis cells are capable of MEMORY operations via Marionette regulators, with concurrent (and orthogonal) T-Pro-regulated CRISPRp of specific *att* sites in a complex memory circuit.

## Engineering a MEMORY-controlled next-generation recombinase-based state machine

To demonstrate how CRISPRp can be used in conjunction with the MEMORY platform for a specific application, we engineered a next-generation recombinase-based state machine (ngRSM) (Fig. 7). In an elegant study, Roquet et al. developed a collection of RSMs in *E. coli*[22]. Briefly, said RSMs used input-driven recombinases to manipulate DNA registers made up of overlapping and orthogonal pairs of recombinase *att* sites via a maximum of three regulated recombinases. DNA registers were designed to adopt a distinct DNA state (predicated on the concurrent recombination of all pairs of cognate *att* sites) for every possible permuted substring of inputs (Supplementary Fig. 13). For example, a 2-input system mapped to a register containing two sets of orthogonal attachment sites – i.e., where recombinase 1 corresponds to an inversion *att* configuration, and recombinase 2 corresponds to two sets of *att* sites distinguished by variation in the central dinucleotide – will result in 5 unique states.

A key limitation of current RSM technology is that DNA registers require complete sets (i.e., even-numbered pairs) of *att* sites to function – such that all *att* sites are recombined in the presence of a cognate recombinase. In principle, a canonical 2-input 5-state RSM can be expanded to a 2-input 16-state RSM via CRISPRp of single *att* sites (Supplementary Fig. 14). Conceptually, the number of inputs that the EcMem chassis can sense and respond to includes the six recombinase inducers, and any additional T-Pro signals corresponding to CRISPRp regulators – e.g., TFs regulating sgRNAs for CRISPRp – including multiple-input T-Pro operations (Fig. 6e). To date, we have developed 5 signal-distinct synthetic repressors[4] and 5 signal-distinct anti-repressors[5,40,67]. In principle, this system of network-capable transcription factors can be used to develop more than 100 2-input T-Pro operations that can be used to regulate and program multiple CRISPRp operations. This creates a rich design space for the development of a vast number of next-generation RSMs where specific recombinases must be induced in the correct order, with CRISPRp providing a method for post-translationally controlling the site of recombinase action.

As a proof-of-concept, we designed, built, and tested a 3-input ngRSM – specifically in the form of a gene-regulatory RSM (GRSM), as defined by Roquet et al. (Fig. 7). As designed, the register of our ngGRSM contains 2 sets of odd-numbered attachment sites (i.e., an *attB/attP* set with a duplicated *att* site) corresponding to Bxb1 and Int3, and one even-numbered set corresponding to Int8. In principle, given the correct sequence of inputs a functional circuit will be formed (recombined) – resulting in the constitutive production of green fluorescent protein. Without CRISPRp this would result in four discrete states; however, if the same register is adapted with CRISPRp this

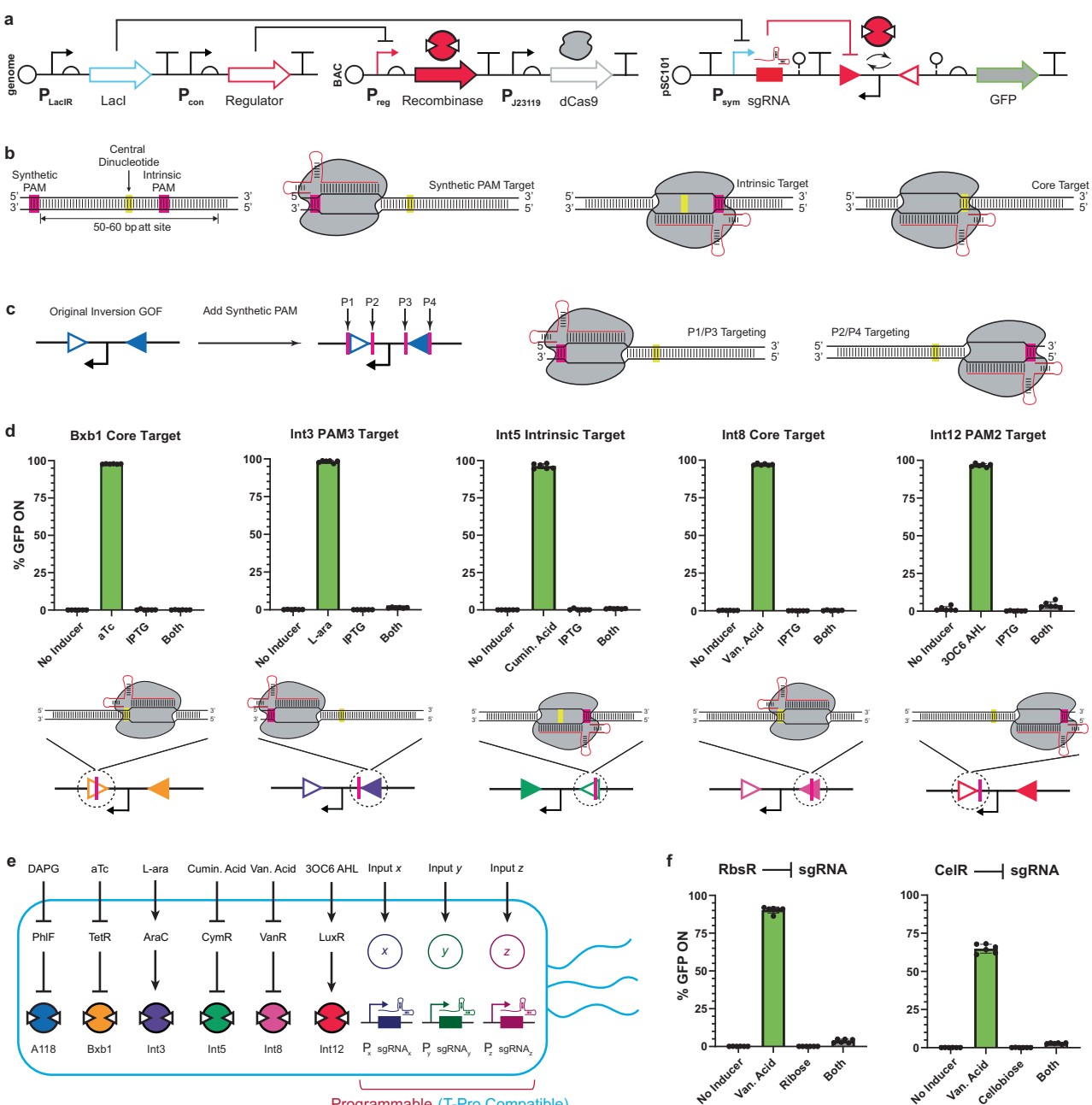

**Fig. 6 | Development of CRISPRp. a** A representative CRISPRp program is shown. LacI regulates sgRNA production while dCas9 is constitutively expressed from the BAC. In the case of no IPTG present in the medium, the recombinase can recombine its target normally when induced. With IPTG in the medium, the sgRNA is produced and directs dCas9 to bind to an *att* site, preventing the recombinase from performing its catalysis. **b** A representative schematic of an *att* site is shown with key features (left), along with a detailed schematic of the putative CRISPRp binding mechanisms (right). **c** An illustration of synthetic PAM addition to the inversion GOF circuit is shown. The positions P1-P4 were assigned based on 5′–3′ directionality, not based on specific *attB/attP* sites. An illustration of the specific strand targeted by each synthetic PAM site is shown to the right. **d** Examples of CRISPRp applied to different recombinases are shown. Marionette-Wild cells transformed with the program described in (**a**) were assayed for recombination with cognate inducer as well as in the presence of IPTG. The specific sgRNA target is shown below each bar graph. **e** A schematic of EcMem with expanded memory capacity is shown. **f** CRISPRp applied to Int8 with sgRNA regulation by RbsR (left) or CelR (right) is shown. Data represent the average of *n* = 6 biological replicates, with groups of three taken on two separate days. Error bars correspond to the SEM of these measurements. Source data are provided as a Source Data file.

would result in nine discrete states (Fig. 7a). The output gene (*gfp*) is initially inaccessible based on a protected Int3 excision GOF circuit. The Int3 excision circuit will only become deprotected if Int8 is first induced, followed by Bxb1. If any MEMORY recombinase is induced out of sequence, a permanent change occurs via nested *att* sites that would otherwise be removed by the correct recombination sequence, permanently preventing the functional circuit from being formed (Fig. 7b). We tested all six possible induction patterns for our ngGRSM

in EcMem and observed near-perfect performance for the prescribed sequences corresponding to their respective expression states (Fig. 7c, d).

**Engineering communication between MEMORY chassis cells**

Having demonstrated synthetic memory and the foundation for decision-making in EcMem chassis cells, we wanted to incorporate tenet 3 (see Fig. 1) of intelligence in the context of MEMORY by

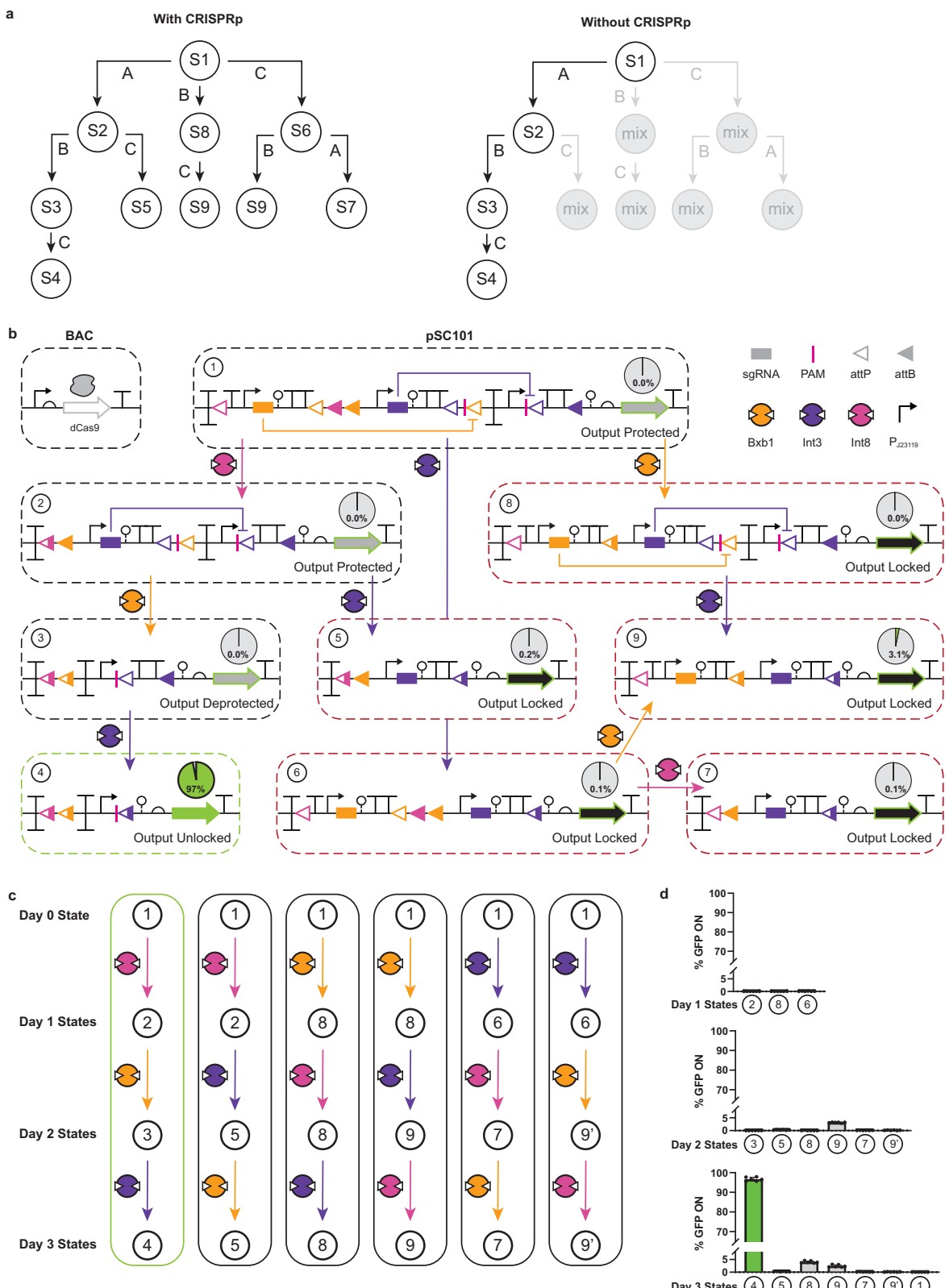

**Fig. 7 | Next-generation recombinase-based state machines. a** A 3-input RSM is shown with 9 possible states based on CRISPRp (left). The same RSM is shown without CRISPRp capability (right). Mix designates nonuniform states. **b** The RSM from **a** is shown as a genetic program. The correct induction pattern (Int8, then Bxb1, then Int3) is required to deprotect and unlock the *gfp* output gene. The EcMem strain transformed with the program was sequentially induced with each relevant inducer in all possible permutations. State transitions are shown by connecting arrows and the state number is shown in the top left of each circled DNA arrangement. The percentage of cells expressing GFP is shown as a pie chart for each program state. **c** Detailed state transitions for (**b**) are shown. The 9′ indicates a synonymous state achieved by a different induction sequence. (**d**) Data from **b** are shown as bar charts representing recombination percentage. Data represent the average of *n* = 6 biological replicates, with groups of three taken on two separate days. Error bars in (**d**) correspond to the SEM of these measurements. Source data are provided as a Source Data file.

including an intercellular communication component. Highlighting that MEMORY inputs DAPG, 3OC6 AHL, and vanillic acid can be synthesized in vivo through their corresponding synthase pathways, we sought to optimize the performance of autoinduction programs that correspond to the cognate biosynthesis pathways for these putative MEMORY communication signals. To this end, we used LacI to regulate the *phlACBD* operon for DAPG production, *luxI* for 3OC6 AHL production, or an *asbF* and *HsOMT* operon for vanillic acid production[68]. Each regulated biosynthetic pathway was designed to function as an inducible input for its cognate biosensor present in the genome-integrated MEMORY system, such that induction would cause a corresponding recombinase to activate an inversion GOF circuit located on the plasmid containing the synthase pathway (Fig. 8a, b). The design goal for each autoinduction program was to achieve near digital recombination performance – i.e., successful recombination of an inversion GOF target with >95% efficiency while exhibiting <5% recombination in the uninduced state. This required synthase pathway tuning via optimization of promoters, RBSs, and degradation tags for each set of genes. In all cases, we were able to achieve the desired autoinduction effect – i.e., near digital recombination performance quantified by flow cytometry (Fig. 8c–e). In turn, we validated these synthase pathways as effective intercellular communication channels by co-culturing EcMem "sender" strains harboring the inducible biosynthetic pathways with EcMem "receiver" strains harboring the appropriate inversion GOF circuits (Fig. 8f–i). Congruent with our design goal, we observed near digital performance for MEMORY-mediated intercellular communication measured by flow cytometry.

### Engineering a probiotic MEMORY strain for therapeutic applications

While the EcMem strain should allow for diverse applications in metabolic engineering and cellular programming, we envisioned that the recombinase-based MEMORY platform could be useful for engineering advanced functionalities into the probiotic *E. coli* Nissle 1917. Therefore, we transferred the MEMORY system into the genome of the Nissle chassis cell to create a probiotic memory strain (EcMem^Pro). We characterized the performance of all 24 recombinase circuits in EcMem^Pro both aerobically and anaerobically (Supplementary Fig. 15), as the typical application of Nissle is in the gastrointestinal (GI) tract. Interestingly, the performance of certain circuits varied unpredictably under anaerobic conditions, but the majority behaved as expected. In principle, the EcMem^Pro strain could be exogenously regulated to produce up to six separate therapeutic modalities by mapping them to orthogonal recombinase circuits.

### Programmed information exchange between EcMem^Pro and *Bacteroides thetaiotaomicron*

Given that Nissle does not stably colonize the human GI tract, it must be administered regularly for it to be effective. *Bacteroides* species, however, are long-term residents of the human colon and are gaining attention as live therapeutic candidates. Having the ability to program information exchange between transient probiotics such as EcMem^Pro and stably colonizing species such as *B. thetaiotaomicron* would represent a paradigm shift in consortium-based living therapeutic technology. To date, there are no reports of synthetic communication systems developed for *Bacteroides* species. To address this challenge, we developed a vanillic acid-inducible circuit in *B. thetaiotaomicron* with Nanoluc as the output (Supplementary Fig. 16). Next, we repurposed our Int12 autoinduction program to produce the vanillic acid biosynthetic pathway as the output instead of GFP (Fig. 8j). We then co-cultured EcMem^Pro cells harboring this program with *B. thetaiotaomicron* cells containing the vanillic acid sensor regulating Nanoluc. The addition of IPTG to the co-culture resulted in a significant increase in Nanoluc expression, at levels comparable to when pure vanillic acid was added (Fig. 8k). This result demonstrates the unification of tenets 1, 2, and 3 in an engineered system, which should facilitate the development of myriad applications in personalized medicine and beyond.

## Discussion

In this study, we engineered intelligent chassis cells capable of concurrent decision-making, memory storage, and intercellular communication by way of a MEMORY platform, significantly advancing the field of synthetic biology. By strategically mapping six optimized biosensors to six orthogonal recombinases, we developed the largest integrated memory circuit platform to date. The optimization of recombinase activity was achieved by the development of near-perfect units of fundamental GOF and LOF operations, which serve as the building blocks for bespoke cellular programs. Specifically, we demonstrated that our MEMORY platform can facilitate programmed inheritable modifications to extrachromosomal DNA and genomic DNA with high efficiency. To complement our synthetic memory system, we introduced a means to post-translationally regulate recombinase action via CRISPRp. In turn, we demonstrated that CRISPRp can be regulated orthogonally using synthetic transcription factors from our Transcriptional Programming toolkit. Complementary to CRISPRp, we recently developed a new form of synthetic memory termed "Interception"[39] (see Supplementary Note 2). Interception is operational with our entire collection of synthetic transcription factors and runs cooperatively with our compressed Boolean logic programs. In principle, CRISPRp can be used concurrently with interception – even on a shared *att* site – which should provide a powerful tool to expand the capabilities of MEMORY chassis cells. Moreover, CRISPRp can be used concurrently with CRISPRi, enabling the coordination of synthetic memory with transient gene knock-down(s), which can be used to increase the versatility of MEMORY chassis cells.

The scale-up of recombinase-based genetic programs has been limited by the number of independently inducible recombinases, as well as the cellular burden they impose when overexpressed from multicopy plasmids[20]. As our engineered EcMem and EcMem^Pro chassis cells double the number of inducible recombinases that can be used in a single *E. coli* cell compared to previous studies[22,30], they have significantly expanded the capacity for recombinase-based programming. Given that the MEMORY system is integrated into the genome and optimized for single-copy performance, our platform technology is operational with minimal metabolic burden. With the array of inducible recombinases modulating DNA-based memory circuits, CRISPRp providing the ability to couple Transcriptional Programming with recombination events, and intercellular communication allowing for multicellular applications, we have provided a platform technology for advanced control over cellular behavior. We demonstrated this capability by programming information exchange between the EcMem^Pro strain and *Bacteroides thetaiotaomicron*, presenting the potential for consortium-based living therapeutic technologies. This system allows for the direct integration of decision-making, memory, and communication in living cells. While we have shown how to apply this platform in the area of living therapeutics, the combined technologies can be used to guide cellular processes in countless ways. We envision that MEMORY strains will be of great use in diverse applications in the areas of metabolic engineering, biosecurity, DNA information storage, and human health.

## Methods

### Bacterial strains and media

*E. coli* strains used were NEB® 10-beta (for routine cloning), TransforMax™ EPI 300™ (for BAC amplification), TransforMax™ EC100D *pir*+ (for R6K plasmid propagation), S17-1 λ pir (for conjugation), MG1655 Marionette-Wild[9], and Nissle 1917 (Mutaflor). *E. coli* were routinely cultured aerobically in LB Miller Medium (Fisher BP9723) at 37 °C (unless otherwise specified) with shaking, on LB Miller agar (Fisher BP1425), or in M9 Minimal Medium (MM) (MM contains 3 g/L $KH_2PO_4$,

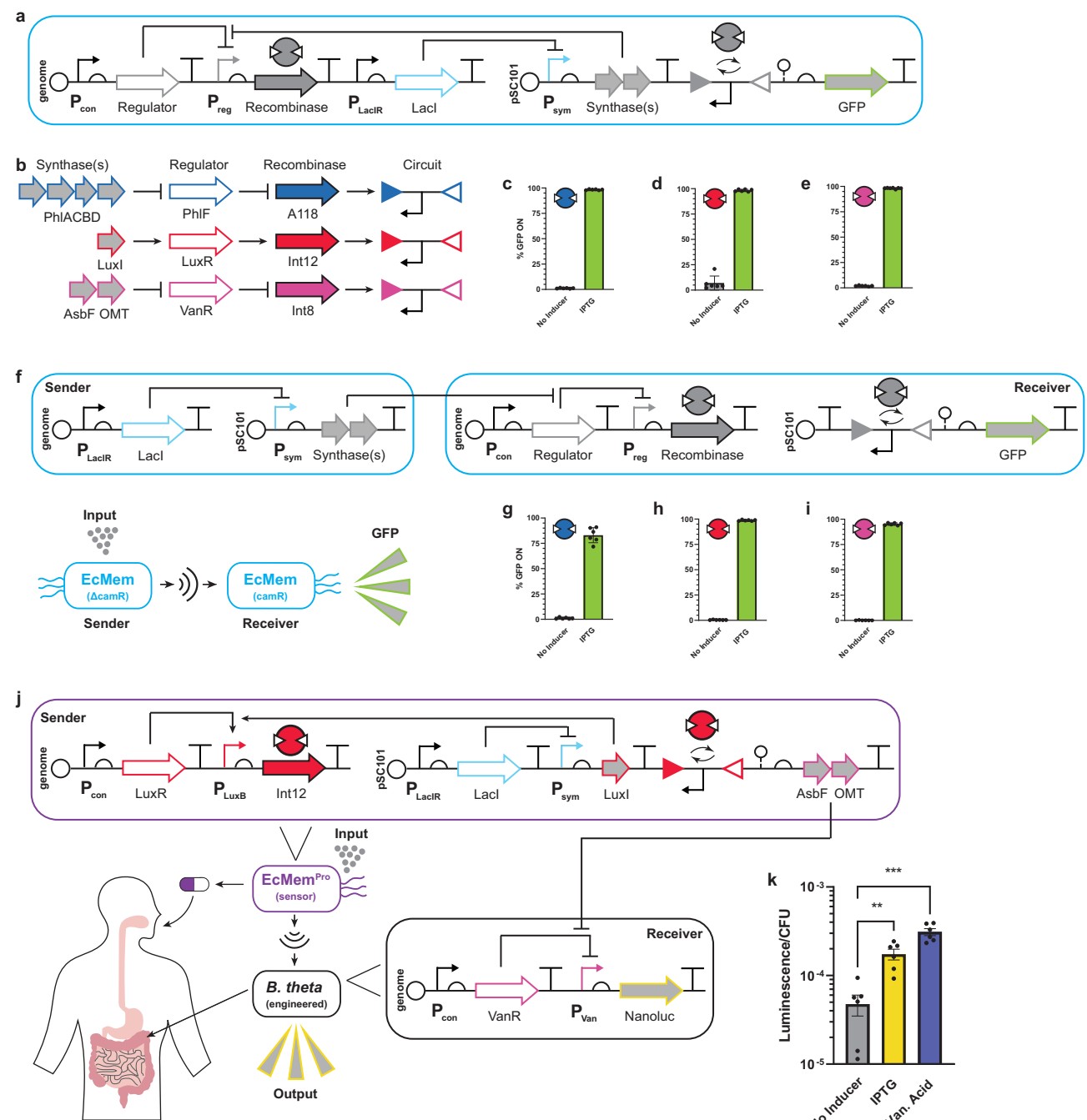

**Fig. 8 | Intercellular communication for programmed information exchange.** **a** A representative autoinduction program is shown. **b** A map of cognate interactions is shown for a synthase pathway to activate a given recombinase. **c** The performance of the A118 autoinduction program is shown for EcMem. **d** The performance of the Int12 autoinduction program is shown for EcMem. **e** The performance of the Int8 autoinduction program is shown for EcMem. **f** A representative intercellular communication program is shown (top) with a conceptual illustration (bottom). The camR label represents the chloramphenicol resistance. **g** Performance of the A118 intercellular communication program is shown for EcMem. **h** Performance of the Int12 intercellular communication program is shown for EcMem. **i** Performance of the Int8 intercellular communication program is

shown for EcMem. **j** A cross-species communication program is shown (top, bottom right) with a conceptual illustration (bottom left). EcMem^Pro harbors an autoinduction program that controls the biosynthesis of vanillic acid while *B. thetaiotaomicron* harbors a vanillic acid-responsive circuit that controls the production of Nanoluc. **k** The luminescence of the EcMem^Pro and *B. thetaiotaomicron* co-culture from (**j**) is shown for different ligand conditions. *P* values are based on unpaired two-tailed *t*-tests with Welch's correction; **P = 2.0E−3; ***P = 6.2E-5. Data represent the average of n = 6 biological replicates, with groups of three taken on two separate days. Error bars correspond to the SEM of these measurements. Source data are provided as a Source Data file.

0.5 g/L NaCl, 6.78 g/L Na₂HPO₄, 1 g/L NH₄Cl, 0.1 mM CaCl₂, 2 mM MgSO₄, 1 mM thiamine hydrochloride, 0.4% D-glucose, and 0.2% casamino acids). *B. thetaiotaomicron* (ATCC 29148) was routinely cultured anaerobically at 37 °C in TYG broth or BHI Agar (Difco), unless otherwise specified. One liter of TYG broth contains: [10 g tryptone, 5 g

yeast extract, 2.5 g D-glucose, 0.5 g L-cysteine, 13.6 g KH₂PO₄, 9.2 mg MgSO₄, 1 g NaHCO₃, 80 mg NaCl, 8 mg CaCl₂, 1 mg menadione, 0.218 mg FeSO₄, 5 μg vitamin B12, and 1 mL histidine hematin solution (1.2 mg/mL hematin in 0.2 M histidine, pH 8.0)]. L-cysteine was resuspended in water and sterile filtered (0.2 μm VWR 28145-477).

Menadione was resuspended in 100% ethanol. L-cysteine and menadione were prepared and added to autoclaved media immediately prior to inoculation. Anaerobic culturing was performed in a Whitley DG250 anaerobic chamber with an atmosphere of 10% $H_2$, 10% $CO_2$, and 80% $N_2$ (Airgas X03NI80C2000511). Antibiotics for plasmid selection in *E. coli* were used at the following concentrations: carbenicillin (Goldbio C-103-25)- 100 µg/mL; chloramphenicol (Goldbio C-105-25)- 25 µg/mL; kanamycin (Goldbio K-120-25)- 35 µg/mL. Antibiotics for *Bacteroides* were used as appropriate: erythromycin (Alfa Aesar J62279)-25 µg/mL and gentamycin (VWR 0304-500 G)-200 µg/mL.

### Chemical inducers

The following chemicals were used as inducers: Isopropyl-beta-D-thiogalactoside (IPTG, Goldbio I2481C); 2,4-Diacetylphloroglucinol (DAPG, Acros Organics 15214288); Anhydrotetracycline HCl (aTc, Alfa Aesar AAJ66688MA); L-arabinose (L-ara, Carbosynth MA02043); Cuminic acid (cuminic acid, Sigma 268402); Vanillic acid (vanillic acid, Alfa Aesar A12074); 3-Oxohexanoyl-homoserine lactone (3OC6 AHL, Sigma K3007); D-Ribose (Alfa Aesar A17894); Cellobiose (Acros Organics 108461000). The final concentrations used for each inducer were: 1 mM IPTG; 25 µM DAPG; 100 ng/mL aTc; 5 mM L-ara; 100 µM cuminic acid; 100 µM vanillic acid; 10 µM 3OC6 AHL; 10 mM Ribose; 10 mM Cellobiose.

### Conjugation of *Bacteroides*

*E. coli* S17-1 λ pir was used for conjugation of plasmids into *Bacteroides*. The pNBU2 vector harbors intN2 which mediates site-specific recombination of the *attN2* site of pNBU2 and one of two *attB2* sites located at the 3' ends of tRNA-Ser genes in *Bacteroides* genomes. Simultaneous insertion of pNBU2 vectors at both sites was not observed, likely due to the necessity of having at least one functional tRNA-Ser gene. Donor cultures of *E. coli* S17-1 λ pir transformed with the appropriate pNBU2 construct and recipient cultures of *Bacteroides* were separately grown to OD600 ~ 0.5. 1 mL of donor culture and 1 mL of recipient culture were pelleted by centrifugation (5000 × *g*, 5 min.) separately and resuspended in 1 mL of PBS. This step was then repeated for a second wash. The cultures were then mixed at a ratio of 1:10 (donor:receiver) and pelleted again by centrifugation. Cells were resuspended in 100 µL PBS and spot plated on a BHI agar plate. The mating lawn was grown aerobically at 37 °C for >16 h before being scraped into 3 mL of PBS. Serial dilutions were plated on BHI agar supplemented with gentamicin and erythromycin. Resultant colonies were picked into TYG after 24–48 h of anaerobic growth. Site-specific integration was confirmed using genome-specific primers.

### Recombinase memory assay

Cells harboring a specific recombinase in the genome or on a BAC were transformed with the desired pSC101 output plasmid and plated on LB agar supplemented with chloramphenicol and kanamycin. After overnight incubation, three colonies were picked into separate 200 µL LB cultures supplemented with chloramphenicol and kanamycin in a flat-bottom 96-well plate (Corning 3370) and sealed with a Breathe Easier membrane (Electron Microscopy Sciences 70536-20). After 8 h of growth in a Thermo Scientific MaxQ 4000 shaker at 300 rpm, these cultures were diluted 1:200 into 200 µL M9 minimal medium with and without the inducer of the specific recombinase. These cultures were sealed with a Breathe Easy membrane (Electron Microscopy Sciences 70536-10) and grown for 12 h, then diluted 1:200 into the fresh medium of the same inducer conditions and grown for an additional 12 h. These cultures were then diluted 1:200 into 200 µL M9 minimal medium containing no inducers and grown for 12–14 h. After this final growth period, the cells were diluted 1:50 into PBS with 2 mg/mL kanamycin to arrest protein production. After greater than 1 h of incubation at room temperature, samples were

processed by flow cytometry to assess recombinase activity (see Cytometry analysis).

### Genomic integration of inducible recombinases – EcMem construction

The inducible recombinase cassettes were serially integrated using the lambda red recombineering method[69]. Briefly, the *a118*, *int12*, *bxb1*, and *int8* genes were cloned into an R6K vector along with a kanamycin resistance cassette, upstream homology to *araE*, and downstream homology to the *glvC* pseudogene. The *int3* and *int5* genes were cloned into a second R6K vector along with a chloramphenicol resistance cassette, upstream homology to *int8*, and downstream homology to the *glvC* pseudogene. BsaI sites were incorporated upstream and downstream of the homology regions to allow for linearization of the DNA to be integrated. Marionette-Wild was transformed with pKD46[69] and made recombineering-ready. Briefly, transformants were selected on LB agar with carbenicillin at 30 °C. A single colony was used to inoculate LB medium with carbenicillin and grown at 30 °C overnight. The following morning this culture was diluted 1:200 into 50 mL fresh LB and grown for 1.5 h. At this point, L-arabinose was added (5 mM) to induce recombineering proteins. Cells were grown for approximately 3 more hours until an OD600 ~ 0.5. Cells were then chilled on ice for 15 min, pelleted by centrifugation (10 min at 4300x *g*), and resuspended in 50 mL ice-cold 10% glycerol. Cells were then pelleted again and resuspended in 25 mL ice-cold 10% glycerol. Cells were pelleted a final time and resuspended in 500 µL ice-cold 10% glycerol. 100 ng of BsaI-linearized DNA (A118, Int12, Bxb1, Int8, and kanR) was electroporated into 50 µL recombineering-ready Marionette-Wild. Transformants were then selected on LB agar with kanamycin and grown at 37 °C. Single colonies were picked into LB medium with kanamycin, grown for 12 h at 37 °C, and then streaked onto fresh LB agar with kanamycin. Resultant colonies were screened for the correct genomic insertion by colony PCR and sequencing of the inserted DNA region. These cells were then made recombineering-ready and the insertion process was repeated with the last two recombinase cassettes, conferring chloramphenicol resistance. This EcMem strain was also modified to remove the chloramphenicol resistance through Flp-mediated excision of the resistance cassette. EcMem^Pro was created in an analogous fashion, but the Marionette transcription factor operons were inserted first. Integration was performed at the LacI locus of EcN. The only difference between the EcMem and EcMem^Pro memory arrays is that the *a118* gene has a GTG start codon in EcMem^Pro.

### Recombinase kinetic assay

The EcMem strain was transformed with each of the six inversion GOF and excision GOF circuits. The following day, individual colonies were used to inoculate 200 µL LB precultures supplemented with chloramphenicol and kanamycin and grown for 10 h in 96-well plates sealed with a Breathe Easier membrane. These cultures were then used to seed MM cultures (1:200 dilution) which were grown for an additional 10 h in 96-well plates sealed with a Breathe Easy membrane. These MM cultures were then used to seed fresh 500 µL MM cultures with the appropriate inducer for a given circuit (1:100 dilution), in deep-well plates (Greiner 780271) sealed with a Breathe Easier membrane. At the same time, an additional set of MM cultures (without inducers) was seeded (1:200 dilution). These uninduced cultures were grown for 12 h and analyzed by flow cytometry to assess recombination levels (designated as the 0-hour time point). The inducing cultures were grown for a total of 16 h. To maintain cells in the exponential phase, the inducing cultures were used to seed fresh inducer-containing media (1:100 dilution) at 8 h. Every 4 h during the induction period, a fresh set of MM cultures without inducers was inoculated using the inducing cultures (1:200 dilution). These inducer-free cultures were all grown for 12 h prior to being analyzed by flow cytometry.

## Genetic stability assay

The EcMem strain was first transformed with the six inversion GOF circuits. Individual colonies were used to inoculate LB medium supplemented with chloramphenicol and kanamycin in 200 μL cultures sealed with a Breathe Easier membrane. These LB precultures were grown for 8 h before being diluted 1:200 into MM with no inducers (sealed with Breathe Easy membranes hereafter). After 12 h of growth, cultures were diluted 1:200 into fresh MM (we designated this as the start of Day 2). Cells were passaged in this manner every 12 h for the remainder of the experiment. Every other day (Days 2, 4, 6, 8, and 10) a separate set of cultures was inoculated from the master set, each containing the inducer corresponding to the recombinase circuit of interest. These inducing cultures were grown for 24 h, with a passaging step at 12 h, and then allowed to grow for 12 h in MM with no inducers. These outgrowth cultures were assayed alongside the uninduced master cultures using flow cytometry. Cells during the final assay had been growing continuously in liquid culture for 11 days.

## Quantification of origin eraser efficacy

After the first day of induction, cells were diluted 1:200 into MM containing L-arabinose and chloramphenicol and grown for an additional 24 h with a passaging step at 12 h. Cells were then diluted 1:200 into MM with chloramphenicol and no inducers and grown for 12 h. After this final growth period, cells were analyzed by flow cytometry to check for pSC101 plasmid loss through GFP fluorescence. These final cultures were also diluted in sterile PBS with no antibiotics over 8 orders of magnitude and spot-plated on LB agar with chloramphenicol only, and with chloramphenicol plus kanamycin. After overnight incubation at 37 °C, colony-forming units (CFU) were counted for each condition.

## CRISPR protection assay

For IPTG-inducible sgRNA CRISPRp circuits, cells transformed with a given circuit were precultured in LB medium supplemented with chloramphenicol and kanamycin with and without IPTG for 8 h in a flat-bottom 96-well plate sealed with a Breathe Easier membrane. After 8 h, the IPTG-free preculture was used to seed MM with and without the cognate inducer of a given recombinase (sealed with a Breathe Easy membrane). The IPTG-containing preculture was used to seed MM with IPTG or IPTG plus the cognate inducer of a given recombinase. These MM cultures were grown for 12 h and then used to seed fresh media (with the same inducer combinations) at a 1:200 dilution. These cultures were grown for an additional 12 h before being diluted 1:200 into MM without any inducers. After a final 12-hour growth period, these cultures were diluted 1:50 into PBS with 2 mg/mL kanamycin for flow cytometry analysis.

## Inducible integration with the memory recorder

The first genomic safe harbor (aGSH1) based on the attP sites of Bxb1 and Int8 was integrated into EcMem through the recombineering method described above. The promoter and attP sites of Bxb1 and Int8 were cloned into the R6K vector along with the kanamycin resistance cassette, upstream homology to the int5 gene, and downstream homology to the glvC pseudogene. The desired insert was digested with BsaI and electroporated into recombineering-ready EcMem cells. Confirmation of genomic insertion was performed as described above. The kanamycin resistance gene was then removed using FLP recombination. EcMem with aGSH1 was transformed with a pSC101 plasmid (equipped with the origin eraser) containing the first gene to be integrated and a kanamycin resistance gene flanked by Int5 and Int12 attP sites (aGSH2), all nested between Bxb1 and Int8 attB sites. Inducible integration was achieved by performing the Memory Assay in MM with kanamycin and inducing cells for 24 h with aTc and vanillic acid. A second 24-hour induction with L-arabinose in the absence of kanamycin was performed to erase the pSC101 plasmid. After each induction period, cells were diluted 1:50 into PBS with 2 mg/mL kanamycin

for cytometry analysis. After L-arabinose induction, cells were streaked on LB agar with 5% sucrose and kanamycin and grown at 37 °C. Correct genomic insertion and pSC101 deletion were confirmed with colony PCR. A single integrated colony was picked and made chemically competent so that a second inducible integration could be performed. These cells were transformed with a new pSC101 plasmid containing the second gene to be integrated and an ampicillin resistance gene (nested between new Bxb1 and Int8 attP sites), all nested between Int5 and Int12 attB sites. These cells were induced in the same manner as the first round, with a 24-hour growth period in MM with cuminic acid plus 3OC6 AHL and carbenicillin, followed by a 24-hour growth period in MM with L-arabinose and no carbenicillin. After each induction period, cells were diluted 1:50 into PBS with 2 mg/mL kanamycin for cytometry analysis. After L-arabinose induction, cells were streaked on LB agar with 5% sucrose and carbenicillin and grown at 37 °C. Correct genomic insertion and pSC101 deletion were confirmed with colony PCR.

## Colony PCR for inducible integration genotyping

Colony PCR was conducted to confirm the insertion of the memory circuit and the deletion of the pSC101 plasmid after inducible integration. After pSC101 deletion with L-arabinose, cells were streaked on LB agar plates and individual colonies were randomly selected for colony PCR. Each colony was diluted in 100 μL DI H$_2$O and 1 μL was added directly to the PCR reaction as a template. One set of primers was designed to specifically bind upstream and downstream of the inserted region to determine if the integration worked correctly. A second set of primers was designed to check the presence of pSC101, testing the deletion of residual DNA sequences. The colony PCR reaction was performed with Q5 polymerase. A 7-minute incubation at 98 °C (5 min for lysis, and 2 min for denaturation of DNA) was followed by a 30-second annealing step, a 30-second extension step, and a 30-second denaturation step (25 cycles). After the PCR, gel electrophoresis was performed with 1:12 diluted PCR products on 0.8% agarose gel with 1 kb DNA Ladder (NEB #N3232). The gel bands were imaged using a ChemiDoc XRS+ System (Bio-Rad) and analyzed using Image Lab software (version 6.0.1). In the first integration colony PCR, the primers named CP1INT_fwd and CP1INT_rev bind to Bxb1 attP and Int8 attP, respectively, checking the length of integration. Primers CP1ERA_fwd and CP1ERA_rev bind to Bxb1 attB and Int3 attB, respectively, checking the presence of pSC101 plasmid. In the second integration colony PCR, the primers named CP2INT_fwd and CP2INT_rev bind downstream of the int5 gene and downstream of glvC gene, respectively, checking the whole length of the insertion. Primers CP2ERA_fwd and CP2ERA_rev bind to the kanR gene, linearizing the pSC101 plasmid. The sequences of colony PCR primers are listed in Supplementary Table 1.

## E. coli intercellular communication assay

Sender cells (lacking chloramphenicol resistance) were transformed with the appropriate synthase plasmid while receiver cells (having chloramphenicol resistance) were transformed with the appropriate inversion GOF circuit. Individual 200 μL LB precultures were inoculated for the sender and receiver cells using single colonies. These cultures were grown for 8 h in a flat-bottom 96-well plate sealed with a Breathe Easier membrane. Following this, the receiver cells were diluted with fresh MM to the following degrees: 1:100 for the A118 and Int12 circuits, and 1:20 for the Int8 circuit. Next, 2.5 μL of the diluted receiver cells and 2.5 μL of the precultured sender cells were seeded together into 1 mL of fresh MM with kanamycin, with and without 1 mM IPTG in deep-well plates (Greiner 780271) sealed with a Breathe Easier membrane. These cultures were grown for 12 h and then diluted 1:200 into 200 μL of fresh MM with kanamycin (with the same IPTG conditions). After 12 h of growth, cultures were diluted 1:200 into fresh MM containing both kanamycin and chloramphenicol (to kill the sender cells). After a final 12 h growth period, cells were prepared for cytometry analysis.

## Co-culture of EcMem^Pro and B. thetaiotaomicron

A single colony of *B. thetaiotaomicron* (harboring the VanR-Nanoluc circuit) was used to inoculate 1 mL TYG and grown anaerobically overnight for 16 h. Meanwhile, EcMem^Pro was transformed with the autoinduction-quorum sensing program. The next day, a single colony of EcMem^Pro was used to inoculate 200 μL LB with kanamycin (to be grown aerobically) while the *B. thetaiotaomicron* culture was diluted 1:200 into fresh TYG. These cultures were grown for 8 h and the *B. thetaiotaomicron* cells were diluted 1:10 with fresh TYG. 5 μL of the diluted *B. thetaiotaomicron* cells and 5 μL of the precultured EcMem^Pro cells were then used to seed 1 mL of TYG cultures without ligand, with IPTG, or with vanillic acid. These cultures were grown anaerobically for 16 h, and then gently mixed with pipetting. 500 μL of cells was then pelleted by centrifugation (10,000 × g for 5 min) and the supernatant was carefully aspirated. The cell pellet was then resuspended in 30 μL Bugbuster Mastermix (Millipore 71456) and incubated at room temperature for 15 min to facilitate cell lysis. Nanoluc production was then quantified using a luminescence assay.

## Luminescence assay

The Promega Nano-Glo assay kit was used to determine expression of NanoLuc. Assay buffer and substrate were mixed as per the manufacturer recommendation (1:50 ratio of substrate to buffer). 30 μL of this mixture was transferred to a well of a flat-bottom white 96-well microplate (Costar 3912) containing 40 μL DI water. Following cell lysis, 30 μL of lysate was added to the microplate well and mixed by pipetting. After 5 min of incubation, the luminescence was measured with a Spectramax M2e plate reader (Molecular Devices) with 800 v gain and 30 reads per well. Data was collected with SoftMax Pro Software (version 7.0.3). Background luminescence generated from an equal mix of EcMem^Pro and wildtype *B. thetaiotaomicron* cells lysed with Bugbuster was subtracted from each sample. Luminescence was then normalized to *B. thetaiotaomicron* colony forming units (CFUs). CFUs were determined for each co-culture by serially diluting the samples in sterile PBS and spot plating 5 μL of the lowest 3 dilutions ($10^{-4}$-$10^{-6}$) in triplicate. Dilution plating was done on BHI agar supplemented with erythromycin (to select for *B. thetaiotaomicron*) and gentamycin (to kill EcMem^Pro). Data was analyzed using Microsoft Excel and Graphpad Prism 9.3.1.

## Cytometry analysis

Fluorescence analysis was performed with a Beckman Coulter Cytoflex S flow cytometer. Cells were diluted 1:50 into PBS with 2 mg/mL kanamycin and incubated for at least 1 h at room temperature. Cells were processed at 10–30 ul/min and monitored through the FITC channel for GFP expression and the ECD channel for mKate expression. Events were gated by forward scatter area vs. side scatter area to eliminate debris and then gated by side scatter height vs. side scatter area to discriminate doublets. More than 10,000 events were collected for final analysis using Cytexpert 2.4 software. A representative gating schematic is shown in Supplementary Fig. 17 and representative flow cytometry data are shown in Supplementary Fig. 18.

## Cloning and plasmid construction

The BAC backbone vector was a kind gift from J. J. Collins (MIT) and J. W. Lee (POSTECH). Recombinase genes were synthesized as gene fragments and subcloned using standard molecular biology techniques. All BAC constructs were created using Golden Gate assembly[70]. pSC101 constructs were created using Golden Gate assembly, inverse PCR, and Gibson cloning[71]. Q5 polymerase (NEB M0491L) was used for PCR. T4 DNA ligase (NEB M0202L), BsmBI-v2 (R0739L), and BsaI-HFv2 (NEB R3733L) were used for Golden Gate cloning. NEBuilder HiFi DNA Assembly Master Mix (NEB E2621X) was used for Gibson cloning. All DNA primers were synthesized by Eurofins Genomics. The DNA sequences of all constructs were verified by Sanger sequencing

(Eurofins Genomics) and visualized using ApE v3.0.6. Relevant plasmid maps are given in Supplementary Figs. 19, 20. Plasmid names are summarized in Supplementary Data 3.

## Reporting summary

Further information on research design is available in the Nature Portfolio Reporting Summary linked to this article.

## Data availability

Plasmid sequences are available from Genbank with accession numbers PP125204-PP125275. Materials generated in this study are available upon request. Source data are provided with this paper.

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

## Acknowledgements
This work was supported by National Science Foundation grants EF 2319231; MCB 2123855; GCR/CBET 1934836; MCB 1921061; CBET 1844289; CBET 1804639 and MCB 1747439 all awarded to C.J.W. We would like to thank J.J. Collins (MIT) and J.W. Lee (POSTECH) for the BAC plasmid, and M. Styczynski (Georgia Tech) for recombineering plasmids.

## Author contributions
B.D.H., D.K., and C.J.W. conceived the study and designed the experiments; B.D.H., D.K., and Y.Y. performed the experiments; B.D.H., D.K., Y.Y., and C.J.W analyzed the data; B.D.H., D.K., and C.J.W wrote the paper.

## Competing interests
The authors declare no competing interests.

## Additional information

**Peer review information** : *Nature Communications* thanks Neel Joshi, Chunbo Lou and the other, anonymous, reviewer(s) for their contribution to the peer review of this work. A peer review file is available.

