## [Peer Review File · Nature Communications]

Engineering Intelligent Chassis Cells via Recombinase-Based MEMORY CircuitsReviewers' Comments:

Reviewer #1:

Remarks to the Author:

Huang et al. present a nice piece of work that significantly advances the field of Synthetic genetic circuits. They have successfully developed a sophisticated platform for creating and controlling intelligent memory in living cells, Honestly, all the excellent regulatory parts were based by previous developed integrase, Chris Voigt's Marionette Biosensor array, previous developed intercellular signaling system and dCas9 regulatory platform. The author might make some optimization for the previous developed regulatory parts, but they do not mark where is their contribution and where belong to others.

Major points

- 1, my major concern for this manuscript is too many functional circuits (GOF, LOF, CRISPRp, ngRSM, state machine and others) were constructed in the work but lack of logic.
2. Another problem was the real application for E. coli Niles and Bacteroides Thetaitotaomicron need to add more discussion and interpretation.
3. Additional suggestion is about the dCas9-based parts which just extensively design in Figure 5, but infrequently used in the following Figure 6 and 7. I guess the section could be separated from the manuscript to write another brief paper, or they can move the material about dCas9-based design into supplementary files.

Reviewer #2:

Remarks to the Author:

The manuscript by Huang et al. describes construction of an E. coli strain containing a chromosomally encoded array of 6 serine integrase ('recombinase') genes which can be regulated independently, allowing for regulated site-specific recombination operations on complex assemblies of sites. The authors go on to demonstrate potential usefulness of this bacterial strain for various Synthetic Biology applications.

The manuscript is very well written throughout, in good English with few errors. The figures are likewise well made and clear, so overall one can read and understand the manuscript OK despite the large amount of data. The experimental work all seems to have been very competently carried out and has been thoroughly and properly described.

The manuscript describes an impressive demonstration of the potential power of this type of strategy for advanced in-cell DNA manipulation, but there is not substantial novelty in the methodology or outcomes. Similar multi-recombinase systems have been reported previously, and all the 'applications' described in the latter part of the manuscript have been exemplified in previous work. Nevertheless, the authors here have 'taken things to a new level' by their comprehensive, thorough approach, so I am inclined to favour publication so as to show readers what can be achieved.

Specific comments

1. Abstract and Introduction. The Abstract could be substantially clearer and more accessible to a non-expert reader. The authors should try to improve it in this way as it would enhance the impact of their paper. Similarly the Introduction, especially the first two paragraphs, is heavy on grandiose and potentially confusing terms ('intelligent synthetic biological systems'; 'iteration of biotic intelligence'; etc.). It could be simplified and clarified to reflect better the background and aims of the actual research presented, emphasising how this work advances the field.

2. Throughout the Experimental section, the authors report remarkably high levels of recombination for all six recombinases (for example in line 124, >97%). These efficiencies are significantly higher than others have seen in comparable experiments. One reason might be the exclusive use of fluorescence reporters and cell counting to measure recombination extent. When the recombination target is on a multicopy plasmid, this method of analysis could substantially overestimate the extent of recombination. For example, with a stable copy number of 5, about 50% recombination would give about 97% fluorescent cells in a 'GOF' experiment. It would be quite simple for the authors to provide independent evidence of the extent of recombination by direct analysis of the plasmid DNA, for example by isolating it from the cells by standard minipreps and analysing it by gel electrophoresis or by re-transformation of empty cells with the plasmid DNA and counting colony colours. This could at least be done for a basic set of data to 'calibrate' the fluorescence measurement analysis.

3. Lines 133 - 135. The term 'address' here is potentially confusing.

4. Lines 171 onwards and throughout. The shortening of the names of the inducers (van. acid, cumin. acid) looks odd and seems unnecessary, or else short names could be defined analogous to aTc.

5. Line 240. "Amendable" - should it be "amenable"?

6. Lines 260, 261. "next-generation recombinase-based state machine" sounds pretentious. Just describe what it actually is.

7. Line 294 "tenant"? Should it be "tenet"?

8. Line 335 "intelligent chassis cell". See comment 6. What is intelligent about it?

9. Line 405 (Figure 2 legend, f). "Evolutionary" is inappropriate here.

10. Line 570. Give a reference for plasmid pKD46.

11. Line 664 onwards ("Colony PCR..." section). This bit of the text is confusing. For example, there seem to be multiple primer sequences with the same names Clarify the presentation.

12. Figure 1d. The chemical structures are almost too small to read and might get pixellated.

13. SI Fig. 13 (b). The figure assumes complete recombination when counting the 'allowed states' - clarify this in the legend.

Reviewer #3:

Remarks to the Author:

This paper describes the expansion of orthogonal recombinases that can be used simultaneously in engineered E. coli-based systems to achieve an array of programmable logic functions, including state machines, memory elements, and intercellular communication. The sheer complexity of the genetic designs and corresponding amount of experimental work presented in the paper is very impressive. The data presented in the paper appears to support its conclusions quite comprehensively. The experiments are designed well in order to demonstrate specific logic functions. The authors also do a good job of maintaining a consistent color-coding scheme throughout their figure images, which facilitates comprehension. The use of six orthogonal recombinases operating as effectively as shown seems like an advance in the state-of-the-art for synthetic biology systems of this kind, though my expertise in this area is not deep enough to compare the authors' approach to competing ones. Nevertheless, I understand enough about the field to appreciate that the approaches described in this paper could be of great utility to the synthetic biology community and beyond, especially if the authors

are willing to distribute their strains. My lab would certainly be interested in obtaining the EcMem and EcMem-Pro strains, if they were available!

Overall, I would whole-heartedly support publication, after addressing the minor critiques outlined below. I have two major comments, but I think it should be up to the authors and editor whether they need to drive any revisions of the paper. The paper's hyper-focus on genetic circuits with increasingly complex logic underlying them made it somewhat difficult to read. Perhaps elements in the main text could be presented in a less jargon-y manner to mitigate this. While the impact of having such a rich toolset will undoubtedly help advance synthetic biology, it is somewhat telling that the final demonstration of the circuit's utility revolves around a fairly straightforward sender-receiver cell experiment. Is the toolset so far ahead of its practical deployment that the authors cannot envision an application that would make full use of its richness? The significance of performing the experiment to demonstrate communication between *B. thetaiotaomicron* and *E. coli* Nissle is clear – this could potentially be a way to deliver serial sets of instructions to a permanent colonizer of the human GI tract. However, it does not require the complexity of the six orthogonal integrases. It also ignores other challenges of GI deployment – fluctuating nutrient conditions, increased fitness stringency, the need to engineer and administer not one, but two engineered strains (and associated regulatory challenges), to name a few.

MINOR COMMENTS:

- I am not sure if this comment is for the authors or the editors, but the lack of a file for reviewers with embedded figures and captions made the paper even harder to read.
- Fig 2f – I am not familiar with this plot type. Perhaps it deserves a bit more explanation. What is the meaning of the vertical lines (and gaps) between the filled and unfilled circles? What is the meaning of the horizontal dotted line?
- The reference to “continuous culture” (line 158) is slightly confusing. Even though the repeated passaging is explained in the next sentence, I would think that term implies culture in a biostat.
- Line 294 – I think this should be “tenet”, not “tenant”
- SI Fig. 15 should have the aerobic and anaerobic datasets labeled in the figure image for clarity.
- Several references are made to anaerobic growth of both *B. thetaiotaomicron* and *E. coli* Nissle, but I could not find the nature of the experiment in the methods or results sections. Were these cultured in an anaerobic chamber

- Neel Joshi (Northeastern University)

Subject: *Nature Communications* Publication—*Research Article (Revision)*

Current Tracking Number: NCOMMS-23-49352A

To Dr. Chuanfu Au (or, whom it may concern),

We are very pleased to hear that our manuscript NCOMMS-23-49352A titled “**Engineering Intelligent Chassis Cells via Recombinase-Based MEMORY Circuits**” and corresponding Supplemental Information are of interest for publication as an article in *Nature Communications*. Thank you and the reviewers for giving us the opportunity to submit a revised manuscript. As instructed, we have addressed all of the concerns raised by the reviewers and editor.

Below is a point-by-point response to each of the reviewer’s comments:

Reviewer #1 (Remarks to the Author):

Reviewer 1: Comment 1. Huang et al. present a nice piece of work that significantly advances the field of Synthetic genetic circuits. They have successfully developed a sophisticated platform for creating and controlling intelligent memory in living cells, Honestly, all the excellent regulatory parts were based by previous developed integrase, Chris Voigt’s Marionette Biosensor array, previous developed intercellular signaling system and dCas9 regulatory platform. The author might make some optimization for the previous developed regulatory parts, but they do not mark where is their contribution and where belong to others.

Response to Reviewer 1: Comment 1. First, we would like to thank the reviewer for the very kind compliment and acknowledgment – i.e., “Huang et al. present a nice piece of work that significantly advances the field of Synthetic genetic circuits”. Second, the reviewer is correct in that many parts have been developed by other groups and we have done the best job within our ability (and length constrains of the journal) acknowledging the appropriate foundational studies and articulating our advance over the state-of-the-art throughout the manuscript. Nevertheless, we took the reviewers advice to make improvements to the paper to better mark our contributions compared to the developments of others. Thank you for helping us improve our paper!

Reviewer 1: Comment 2. my major concern for this manuscript is too many functional circuits (GOF, LOF, CRISPRp, ngRSM, state machine and others) were constructed in the work but lack of logic.

Response to Reviewer 1: Comment 2. First, we acknowledge the reviewer’s concern noted as “too many functional circuits”. We felt the need to clearly and overwhelmingly prove our technical ability and win the confidence of the reader as we built to our advance over the state-of-the-art given the complexity of many of the latter circuits. Second, to improve the “logic” of our work as it relates to the aforesaid circuits we have added an additional figure (see Figure 1), expanded the text, and re-wrote the abstract to aid in articulating the narrative in the context of the “tenets” of intelligence - i.e., decision-making (tenet 1), memory (tenet 2), and communication (tenet 3).

Reviewer 1: Comment 3. Another problem was the real application for E. coli Niles and Bacteroides Thetaiotaomicron need to add more discussion and interpretation.

Response to Reviewer 1: Comment 3. We have added additional explanations of the application of this experiment as suggested. Also note we have published an extensive paper outlining the utility of the engineered Bacteroides chassis cells for use as living therapeutics – see Nature Communications (2022) 13 (1), 3901.

Reviewer 1: Comment 4. Additional suggestion is about the dCas9-based parts which just extensively design in Figure 5, but infrequently used in the following Figure 6 and 7. I guess the section could be separated from the manuscript to write another brief paper, or they can move the material about dCas9-based design into supplementary files.

Response to Reviewer 1: Comment 4. This particular design demonstrates the coupling of decision-making (tenet 1) using Transcriptional Programming and memory (tenet 2) using our recombinase system, which is a critical part of designed intelligence. Accordingly, the inclusion of this section is an important part of the overall narrative. Accordingly, we have modified the text to better reflect this point.

Reviewer #2 (Remarks to the Author):

Reviewer 2: Comment 1. The manuscript by Huang et al. describes construction of an E. coli strain containing a chromosomally encoded array of 6 serine integrase ('recombinase') genes which can be regulated independently, allowing for regulated site-specific recombination operations on complex assemblies of sites. The authors go on to demonstrate potential usefulness of this bacterial strain for various Synthetic Biology applications.

The manuscript is very well written throughout, in good English with few errors. The figures are likewise well made and clear, so overall one can read and understand the manuscript OK despite the large amount of data. The experimental work all seems to have been very competently carried out and has been thoroughly and properly described.

The manuscript describes an impressive demonstration of the potential power of this type of strategy for advanced in-cell DNA manipulation, but there is not substantial novelty in the methodology or outcomes. Similar multi-recombinase systems have been reported previously, and all the 'applications' described in the latter part of the manuscript have been exemplified in previous work. Nevertheless, the authors here have 'taken things to a new level' by their comprehensive, thorough approach, so I am inclined to favour publication so as to show readers what can be achieved.

Response to Reviewer 2: Comment 1. We would like to thank the reviewer for their time and consideration when reading our manuscript and favoring publication.

Reviewer 2: Comment 2. Abstract and Introduction. The Abstract could be substantially clearer and more accessible to a non-expert reader. The authors should try to improve it in this way as it would enhance the impact of their paper. Similarly the Introduction, especially the first two paragraphs, is heavy on grandiose and potentially confusing terms ('intelligent synthetic biological systems'; 'iteration of biotic intelligence'; etc.). It could be simplified and clarified to reflect better the background and aims of the actual research presented, emphasising how this work advances the field.

Response to Reviewer 2: Comment 2. We have revised the abstract to make it more accessible as instructed by the reviewer. To improve the narrative with regard to intelligence we have added an additional figure (see Fig. 1) and text to improve the narrative in the context of three fundamental tenets - i.e., decision-making (tenet 1), memory (tenet 2), and communication (tenet 3). Likewise, we minimized the use of “potentially confusing terms” beyond (now defined) intelligence and the corresponding tenets. Note, engineered intelligence is one of the central themes of our work, which we have built over several papers culminating in the current (entitled) paper.

Reviewer 2: Comment 3. Throughout the Experimental section, the authors report remarkably high levels of recombination for all six recombinases (for example in line 124, >97%). These efficiencies are significantly higher than others have seen in comparable experiments. One reason might be the exclusive use of fluorescence reporters and cell counting to measure recombination extent. When the recombination target is on a multicopy plasmid, this method of analysis could substantially overestimate the extent of recombination. For example, with a stable copy number of 5, about 50% recombination would give about 97% fluorescent cells in a 'GOF' experiment. It would be quite simple for the authors to provide independent evidence of the extent of recombination by direct analysis of the plasmid DNA, for example by isolating it from the cells by standard minipreps and analysing it by gel electrophoresis or by re-transformation of empty cells with the plasmid DNA and counting colony colours. This could at least be done for a basic set of data to 'calibrate' the fluorescence measurement analysis.

Response to Reviewer 2: Comment 3. As suggested by the reviewer, recombined control plasmids were obtained by miniprepping DNA from the cells that recombined them, and we confirmed that the phenotype measured by flow is consistent with genotype. For example, we see that partial recombination result in a bimodal distribution, and the fluorescence measured by flow is consistent with the plasmid genotypes. This contrasts with cells with near complete (95-99%) recombination confirmed by genotype and phenotype, where distribution is not bimodal. Note it is true that with higher copy numbers – cognate to the attachment sites – the optimized MEMORY will likely be less efficient - thus the MEMORY unit would require tuning. We appreciate the reviewer’s concern regarding the potential overestimation of recombinase efficiencies. We agree that we achieved very high levels of recombination, but this is comparable to papers that used flow cytometry and genotyping to assess recombination levels¹⁻³ (routinely reporting 95-99% recombination). When using cytometry to count cells, we based our gates on unrecombined and recombined control plasmids (See SI Fig. 18). Additionally, we have further evidence that our efficiency measured by cytometry is accurate based on the plasmid erasure experiment in Fig. 4 and SI Figs. 7-8. We can see here that the cells deemed GFP negative (lost the plasmid) is comparable to the survival ratio when plated on selective and non-selective media.

Reviewer 2: Comment 4. Lines 133 - 135. The term 'address' here is potentially confusing.

Response to Reviewer 2: Comment 4. We have changed address to circuit (or to a less confusing term) throughout the manuscript as appropriate.

Reviewer 2: Comment 5 Lines 171 onwards and throughout. The shortening of the names of the inducers (van. acid, cumini. acid) looks odd and seems unnecessary, or else short names could be defined analogous to aTc.

Response to Reviewer 2: Comment 5. We have removed the abbreviations for vanillic acid, cuminic acid, and L-arabinose and replaced them with the full words in the main text.

Reviewer 2: Comment 6. Line 240. "Amendable" - should it be "amenable"?

Response to Reviewer 2: Comment 6. Yes – this has now been corrected.

Reviewer 2: Comment 7. Lines 260, 261. "next-generation recombinase-based state machine" sounds pretentious. Just describe what it actually is.

Response to Reviewer 2: Comment 7. We appreciate the Reviewer's comment. The term "recombinase-based state machine" has already been defined by Roquet et al.; accordingly, to maintain consistency with previously published work we opted to keep the naming convention. In addition, calling our technology a "next-generation" recombinase-based state machine seems appropriate to emphasize an important advance and marking a different iteration of this technology – given that the term "next-generation" is commonly used to make such distinctions. Please pardon us if we did not fully appreciate the reviewer's intention with this comment.

Reviewer 2: Comment 8. Line 294 "tenant"? Should it be "tenet"?

Response to Reviewer 2: Comment 8. Yes – this has now been corrected.

Reviewer 2: Comment 9. Line 335 "intelligent chassis cell". See comment 6. What is intelligent about it?

Response to Reviewer 2: Comment 9. In order to better clarify what we mean with regard to an "intelligent chassis cell" we have included an additional figure (see Fig. 1) and text.

Reviewer 2: Comment 10. Line 405 (Figure 2 legend, f). "Evolutionary" is inappropriate here.

Response to Reviewer 2: Comment 10. We have changed the term "evolutionary stability" to "genetic stability".

Reviewer 2: Comment 11. Line 570. Give a reference for plasmid pKD46.

Response to Reviewer 2: Comment 11. Done

Reviewer 2: Comment 12. Line 664 onwards ("Colony PCR..." section). This bit of the text is confusing. For example, there seem to be multiple primer sequences with the same names Clarify the presentation.

Response to Reviewer 2: Comment 12. Done, we have re-written this section to improve clarity.

Reviewer 2: Comment 13. Figure 1d. The chemical structures are almost too small to read and might get pixellated.

Response to Reviewer 2: Comment 13. Noted – we have removed the chemical structures for this figure.

Reviewer 2: Comment 14. SI Fig. 13 (b). The figure assumes complete recombination when counting the 'allowed states' - clarify this in the legend.

Response to Reviewer 2: Comment 14. Done, we have added this clarification.

Reviewer #3 (Remarks to the Author):

Reviewer 3: Comment 1. This paper describes the expansion of orthogonal recombinases that can be used simultaneously in engineered *E. coli*-based systems to achieve an array of programmable logic functions, including state machines, memory elements, and intercellular communication. The sheer complexity of the genetic designs and corresponding amount of experimental work presented in the paper is very impressive. The data presented in the paper appears to support its conclusions quite comprehensively. The experiments are designed well in order to demonstrate specific logic functions. The authors also do a good job of maintaining a consistent color-coding scheme throughout their figure images, which facilitates comprehension. The use of six orthogonal recombinases operating as effectively as shown seems like an advance in the state-of-the-art for synthetic biology systems of this kind, though my expertise in this area is not deep enough to compare the authors' approach to competing ones. Nevertheless, I understand enough about the field to appreciate that the approaches described in this paper could be of great utility to the synthetic biology community and beyond, especially if the authors are willing to distribute their strains. My lab would certainly be interested in obtaining the EcMem and EcMem-Pro strains, if they were available!

Response to Reviewer 3: Comment 1. We would like to thank the reviewer for the very kind remarks and careful reading of the manuscript. Of course, we would be delighted to share any of the chassis cells and any related DNA components with the reviewer! Please note we will make all technologies developed in this manuscript available via public repository.

Reviewer 3: Comment 2. Overall, I would whole-heartedly support publication, after addressing the minor critiques outlined below. I have two major comments, but I think it should be up to the authors and editor whether they need to drive any revisions of the paper. The paper's hyper-focus on genetic circuits with increasingly complex logic underlying them made it somewhat difficult to read. Perhaps elements in the main text could be presented in a less jargon-y manner to mitigate this. While the impact of having such a rich toolset will undoubtedly help advance synthetic biology, it is somewhat telling that the final demonstration of the circuit's utility revolves around a fairly straightforward sender-receiver cell experiment. Is the toolset so far ahead of its practical deployment that the authors cannot envision an application that would make full use of its richness? The significance of performing the experiment to demonstrate communication between *B. theta* and *E. coli* Nissle is clear – this could potentially be a way to deliver serial sets of instructions to a permanent colonizer of the human GI tract. However, it does not require the complexity of the six orthogonal integrases. It also ignores other challenges of GI deployment – fluctuating nutrient conditions, increased fitness stringency, the need to engineer and administer not one, but two engineered strains (and associated regulatory challenges), to name a few.

Response to Reviewer 3: Comment 2. First, we would like to thank the reviewer for “whole-heartedly support[ing] publication”. We appreciate the comment about the readability of the paper. We have attempted to reduce jargon and confusing language throughout as requested. We focused our efforts on refining the toolkits presented in the manuscript so that they could be used in broad applications, many of which we envision as entire new projects that we are currently pursuing. We agree that the EcN-*B. theta* experiment does not require the whole MEMORY platform, but it does utilize multiple technologies we developed in the paper working in concert – i.e., decision-making (tenet 1), memory (tenet 2), and communication (tenet 3) – which we have now summarized in a Figure to guide the discussion. This experiment was designed as a proof of concept, and we agree that real-world deployment may be more difficult. However, techniques for administering and engrafting *Bacteroides* have been developed⁴, supporting the possibility of this concept. The administration of EcN would then be trivial via an oral pill since it is intended to be a transient strain in the body.

MINOR COMMENTS:

Reviewer 3: Comment 3. I am not sure if this comment is for the authors or the editors, but the lack of a file for reviewers with embedded figures and captions made the paper even harder to read.

Response to Reviewer 3: Comment 3. I believe the fault belongs to us, and we apologize for any undue burden.

Reviewer 3: Comment 4. Fig 2f – I am not familiar with this plot type. Perhaps it deserves a bit more explanation. What is the meaning of the vertical lines (and gaps) between the filled and unfilled circles? What is the meaning of the horizontal dotted line?

Response to Reviewer 3: Comment 4. We have edited this plot for clarity. The dots represent recombination of uninduced vs induced cultures, with the line connecting the two for a single recombinase – now given as a legend to the left of the plot. The horizontal line was to guide the eye to the very low value based on the split Y-axis, but we removed it for clarity.

Reviewer 3: Comment 5. The reference to “continuous culture” (line 158) is slightly confusing. Even though the repeated passaging is explained in the next sentence, I would think that term implies culture in a biostat.

Response to Reviewer 3: Comment 5. We agree with the comment. We have removed the reference to continuous culture for clarity.

Reviewer 3: Comment 6. Line 294 – I think this should be “tenet”, not “tenant”

Response to Reviewer 3: Comment 6. Thank you, this was corrected.

Reviewer 3: Comment 7. SI Fig. 15 should have the aerobic and anaerobic datasets labeled in the figure image for clarity.

Response to Reviewer 3: Comment 7. The labels have been moved to the top of the figure for clarity.

Reviewer 3: Comment 8. Several references are made to anaerobic growth of both *B. thetaiotaomicron* and *E. coli* Nissle, but I could not find the nature of the experiment in the methods or results sections. Were these cultured in an anaerobic chamber

Response to Reviewer 3: Comment 8. We have clarified in the Methods section how anaerobic growth was performed (DG250 anaerobic chamber). The details of media and co-culture procedures are in the Methods as well.

Comments to the Editor:

In addition to the above we have reviewed and completed the relevant check lists, and have made all necessary changes to comply with *Nature Communications* publication standards - to the best of our knowledge. Below is a summary of completed tasks:

1. Nature Communications Manuscript Check List
2. Editorial policy checklist (Returned)
3. Reporting requirements for life sciences research (Returned)
4. Data availability statements and data citations policy

Please let me know if you need any additional information.

S

Reviewers' Comments:

Reviewer #1:

Remarks to the Author:

The authors have addressed all my concerns

Reviewer #2:

Remarks to the Author:

As far as I can tell the authors have satisfactorily addressed all the points in the original referees' reports, so I am happy to recommend that the manuscript should now be published. I have just a few minor points (see below) that I noticed on reading the revised manuscript, which might improve the paper in its final version.

(1) Line 107. I don't think the authors mean "...if its coding sequence was in frame with..." - the text here seems to be discussing the possibility of transcriptional readthrough, not translation.

(2) Line 119. "the MG1655 genome." Clarify here that the strain is the "Marionette" derivative of MG1655?

(3) Lines 393 and 398. Clarify that coding sequences for GFP or recombinases are downstream of the promoter/terminator, not the proteins themselves.

(4) Line 658. Greek letter mu, not u (twice).

(5) Figure 4, part c. It might help the reader to add a label to the figure or state in the legend that the top line of part c refers to the data shown in parts a and b above (i.e. "Execute program 1; Erase program1).

Reviewer #3:

Remarks to the Author:

I think the authors have done a good job on the revision. All of my critiques have been addressed satisfactorily. The critiques from other reviewers also seem to have been addressed. In my opinion, the revised manuscript is stronger, reads better, and is fit for publication.

Subject: *Nature Communications* Publication—*Research Article (Revision)*

Current Tracking Number: NCOMMS-23-49352A

To Dr. Chuanfu Au (or, whom it may concern),

We are very pleased to hear that our manuscript NCOMMS-23-49352A titled “**Engineering Intelligent Chassis Cells via Recombinase-Based MEMORY Circuits**” and corresponding Supplemental Information are of interest for publication as an article in *Nature Communications*. Thank you and the reviewers for giving us the opportunity to submit a revised manuscript. As instructed, we have addressed all of the concerns raised by the reviewers and editor.

Below is a point-by-point response to each of the reviewer’s comments:

Reviewer #1 (Remarks to the Author):

Reviewer 1 Comment 1: The authors have addressed all my concerns

Response: We would like to thank the Reviewer for their helpful feedback.

Reviewer #2 (Remarks to the Author):

Reviewer 2 Comment 1: As far as I can tell the authors have satisfactorily addressed all the points in the original referees' reports, so I am happy to recommend that the manuscript should now be published. I have just a few minor points (see below) that I noticed on reading the revised manuscript, which might improve the paper in its final version.

Response: We would like to thank the Reviewer for their critical reading of the manuscript and helpful comments for improving textual clarity.

Reviewer 2 Comment 2: Line 107. I don't think the authors mean "...if its coding sequence was in frame with..." - the text here seems to be discussing the possibility of transcriptional readthrough, not translation.

Response: We have revised this sentence to clarify that we are discussing transcriptional readthrough.

Reviewer 2 Comment 3: Line 119. "the MG1655 genome." Clarify here that the strain is the "Marionette" derivative of MG1655?

Response: We have added this clarification.

Reviewer 2 Comment 4: Lines 393 and 398. Clarify that coding sequences for GFP or recombinases are downstream of the promoter/terminator, not the proteins themselves.

Response: We have clarified that we are referring to DNA sequences.

Reviewer 2 Comment 5: Line 658. Greek letter mu, not u (twice).

Response: This has been corrected.

Reviewer 2 Comment 6: Figure 4, part c. It might help the reader to add a label to the figure or state in the legend that the top line of part c refers to the data shown in parts a and b above (i.e. "Execute program 1; Erase program1).

Response: We have added these additional details to the legend as suggested.

Reviewer #3 (Remarks to the Author):

Reviewer 2 Comment 1: I think the authors have done a good job on the revision. All of my critiques have been addressed satisfactorily. The critiques from other reviewers also seem to have been addressed. In my opinion, the revised manuscript is stronger, reads better, and is fit for publication.

Response: We thank the reviewer for their time and support.

Comments to the Editor:

In addition to the above we have reviewed and completed the relevant check lists, and have made all necessary changes to comply with *Nature Communications* publication standards - to the best of our knowledge. Below is a summary of completed tasks:

1. Nature Communications Manuscript Check List
2. Editorial policy checklist (Returned)
3. Reporting requirements for life sciences research (Returned)
4. Data availability statements and data citations policy

Please let me know if you need any additional information.